# Learning Scenario Representation for Solving Two-stage Stochastic Integer Programs

**Yaoxin Wu[1][*], Wen Song[2][*], Zhiguang Cao[3][†], Jie Zhang[1]**

[1]School of Computer Science and Engineering, Nanyang Technological University, Singapore
[2]Institute of Marine Science and Technology, Shandong University, Qingdao, China
[3]Singapore Institute of Manufacturing Technology, A*STAR, Singapore
`{wuyaoxin,zhangj}@ntu.edu.sg; wensong@email.sdu.edu.cn`

## Abstract

Many practical combinatorial optimization problems under uncertainty can be modeled as stochastic integer programs (SIPs), which are extremely challenging to solve due to the high complexity. To solve two-stage SIPs efficiently, we propose a conditional variational autoencoder (CVAE) based method to learn scenario representation for a class of SIP instances. Specifically, we design a graph convolutional network based encoder to embed each scenario with the deterministic part of its instance (i.e. context) into a low-dimensional latent space, from which a decoder reconstructs the scenario from its latent representation conditioned on the context. Such a design effectively captures the dependencies of the scenarios on their corresponding instances. We apply the trained encoder to two tasks in typical SIP solving, i.e. scenario reduction and objective prediction. Experiments on two graph-based SIPs show that the learned representation significantly boosts the solving performance to attain high-quality solutions in short computational time, and generalizes fairly well to problems of larger sizes or with more scenarios.

## 1 Introduction

Stochastic integer programs (SIPs) are a class of combinatorial optimization problems (COPs) with uncertain parameters. They focus on finding a best decision for a practical task to optimize both the deterministic cost and the expectation of stochastic costs on a set of potential *scenarios*. Since the uncertain elements commonly exist in reality, SIPs are widely applied and studied in many fields such as transportation, inventory management, energy optimization, and so on (Jammeli et al., 2019; Dillon et al., 2017; Bruno et al., 2016; Wallace & Ziemba, 2005). However, optimally solving SIP is intractable especially when many scenarios are involved, due to the fact that: 1) most integer programs themselves are already NP-hard; 2) considerable (even continuous) scenarios significantly increase the computational complexity (Klein, 2021). In reality, solving moderate-sized problems may require prohibitively long time. Thus efficient and high-quality approximate solutions for SIPs are often practically pursued.

The advances in deep (reinforcement) learning has aroused extensive studies on solving optimization problems with neural networks. The related works have shown promising results in tackling some classic COPs, e.g., vehicle routing, job shop scheduling, combinatorial auction, etc. (Kool et al., 2019; Zhang et al., 2020; Lee et al., 2020; Kwon et al., 2020; Gasse et al., 2019; Chen & Tian, 2019; Ichter et al., 2018; Hottung et al., 2020; Wu et al., 2021). However, most of the existing deep learning based methods only focus on deterministic problems, which could hinder their practical applications in the uncertain situation. On the other hand, a few early attempts introduce deep learning to solve SIPs (Larsen et al., 2018; Abbasi et al., 2020; Joe & Lau, 2020; Bengio et al., 2020; Nair et al., 2018). However, they are generally designed for only specific problems and also lack the ability of generalizing across different problem sizes.

In this paper, we focus on graph-based SIPs, which is a general class of problems with broad applications. We propose to learn the deep latent representation of *scenarios* in SIPs, which can be

---

[*]Equal contribution.
[†]Corresponding author (`zhiguangcao@outlook.com`).

applied to various downstream tasks in solving SIPs, e.g., *scenario reduction* and *objective prediction*. Based on the basic two-stage form of SIPs, we leverage conditional variational autoencoder (CVAE) to learn latent and continuous representations of stochastic scenarios while considering the deterministic part of a SIP instance, which we term the *context*. Specifically, we employ a graph convolutional network (GCN) to learn embeddings of each deterministic integer program, which models the first-stage problem with a single scenario. The learned embeddings are mapped into continuous representations, and then decoded back into corresponding scenarios given the embedding of the context. This design ensures that the learned representations of scenarios are correctly linked to the corresponding instances.

The resulting latent space well represents scenarios in a class of SIP instances, and we demonstrate its power with two tasks commonly used in solving SIPs. First, we leverage it for scenario reduction by finding representative scenarios via an off-the-shelf clustering algorithm, which decreases the problem complexity and meanwhile attains high-quality approximate solutions. Second, we further extend the encoder with an additional sub-network to predict objective values for each scenario. Due to the generalization in the latent continuous space, we readily deploy the prediction task in semi-supervised manner with a small number of target objective values. In doing so, the latent space is augmented with the predicted objective value and hence yields more representative scenarios, which further shrinks the approximation gap to the optimal solutions of the original SIPs.

In summary, this paper aims to learn scenario representation for solving graph-based two-stage SIPs. The main contributions are as follows:

- We firstly present a deep generative framework to learn representations for scenarios in graph-based two-stage SIPs. It considers dependencies of the scenarios on the deterministic context of each instance, and hence well differentiates scenarios in the latent space.

- We apply the learned representations to scenario reduction and objective prediction for SIPs. The representative scenarios derived from clustering of the representations can be used to efficiently compute high-quality solutions for SIPs. Moreover, we further shrink the approximation error by predicting objective values of scenarios via semi-supervised learning.

- We evaluate the proposed method on two SIP problems, i.e., the network design problem and facility location problem. Experimental results verify the superiority of the learned scenario representation for solving SIPs. Our method significantly outperforms existing deep learning based baselines. Notably, it also generalizes well to problems with larger sizes or more scenarios.

## 2 RELATED WORK

**Learning based methods for SIPs.** Recent works of deep learning for COPs mostly address deterministic problems, with scarce attention to uncertainty that commonly exist in practice. Though a few methods apply deep learning to improve approximate solutions for two-stage SIPs, they are mostly designed for some specific problems. Nair et al. (2018) train a bit-flipping policy with reinforcement learning to improve solutions iteratively, but it is limited to binary decision variables and assumes no constraints in the first-stage problem. Larsen et al. (2018) predict properties of optimal solutions to integer programs (i.e. the first-stage problem with respective single scenario) in a supervised manner. This method is only applicable when operational solutions are not needed. In contrast, Abbasi et al. (2020) predict values of decision variables for blood transshipment problem with fixed-sized instances. To reduce the complexity of SIPs, Bengio et al. (2020) aim to yield one scenario for the first-stage problem, so that the derived solution achieves a small objective value when evaluated with the original set of scenarios. This method assumes complete recourse, with any generated scenario being feasible for the second-stage problem. All the above methods attempt to reduce the expected cost of stochastic scenarios but their applications are limited by distinct assumptions or fixed problem sizes. In contrast, our method can be used for more general two-stage SIPs, and generalizes well to varying problem sizes or scenario quantities.

**Scenario reduction for SIPs.** In the SIP formulation, a set of scenarios is used to model stochastic events on top of the first-stage problem. It introduces extra complexity which may exponentially increase with its cardinality, owing to the combinatorial search space (Dyer & Stougie, 2006). Since the direct optimization with massive scenarios is intractable, the scenario reduction is often conducted to find a subset of scenarios that well replaces the original set with a small approximation

error. Existing reduction methods mainly branches into distribution-oriented and problem-oriented paradigms. The former ones pursue a group of scenarios that closely estimate the probability distribution under the scenario space (Henrion & Römisch, 2017). The distance between the estimated and actual probability are often measured by Wasserstein distance (Rujeerapaiboon et al., 2018), polyhedral discrepancy (Henrion et al., 2008) or difference of moments (Høyland et al., 2003). These methods neglect the influence of the context on the objective function, e.g., similar scenario distributions may cause different expected costs in disparate SIP instances. Thus, there is a recent trend towards the study of problem-oriented methods (Henrion & Römisch, 2018; Fairbrother et al., 2019; Keutchayan et al., 2020; 2021), which attempts to involve problem-specific properties in scenario reduction. For example, Keutchayan et al. (2021) propose to find $K$ scenarios with their objective values separately approximating the expected objective values in the $K$ scenario subsets. Most the above methods are mainly for theoretical proof on stochastic programs (without the restriction of integers). In contrast, we firstly propose a deep learning method for SIPs to learn scenario representation, which can be used for both scenario reduction and objective prediction.

## 3 PRELIMINARIES

### 3.1 TWO-STAGE STOCHASTIC INTEGER PROGRAMS

As an optimization problem, SIP is usually characterized by both the uncertain parameters that potentially follow certain probability distributions and the discrete solution space due to the integer restrictions. It is commonly described by a two-stage formulation as below:

$$\min_{x} \quad \mu^\top x + \mathbb{E}_\omega[Q(x, \omega)] \tag{1}$$

$$\text{s.t.} \quad Ax \leq b, \ x \in \mathbb{R}^{n_1 - p_1} \times \mathbb{Z}^{p_1}, \tag{2}$$

where $Q(x, \omega) := \min_y \{q_\omega^\top y | W_\omega y \leq h_\omega - T_\omega x; y \in \mathbb{R}^{n_2 - p_2} \times \mathbb{Z}^{p_2}\}$.

In particular, Eq. (1) and (2) prescribe the first-stage problem with $x \in \mathbb{R}^{n_1}$ being the decision variable, where $n_1 \geq p_1$; $Q(x, \omega)$ prescribes the second-stage problem, with $y \in \mathbb{R}^{n_2}$ being the decision variable, where $n_2 \geq p_2$. We refer to the group of *static* parameters $\mu \in \mathbb{R}^{n_1}$, $A \in \mathbb{R}^{m_1 \times n_1}$ and $b \in \mathbb{R}^{m_1}$ in the first stage as the *context*, and assume that the group of *uncertain* parameters $q_\omega \in \mathbb{R}^{n_2}$, $W_\omega \in \mathbb{R}^{m_2 \times n_2}$, $T_\omega \in \mathbb{R}^{m_2 \times n_1}$ and $h_\omega \in \mathbb{R}^{m_2}$ follows a distribution $\mathcal{P}$. Here we focus on SIPs defined on graphs, which are a family of problems with practical applications in many domains such as networks, transportation and scheduling (Rahmaniani et al., 2018; An & Lo, 2016).

The primary method to solve the above SIP is sample average approximation (Kleywegt et al., 2002), which converts it into a mixed integer program (MIP) by Monte Carlo simulation and in turn optimizes the following objective:

$$\mathcal{O}(x) := \min_{x} \quad \mu^\top x + \frac{1}{N} \sum_{i=1}^{N} Q(x, \omega_i), \tag{3}$$

where $\{\omega_i\}_{i=1}^N$ is a set of *scenarios* from $\mathcal{P}$, i.e., an independently and identically distributed (i.i.d.) random sample of $N$ realizations of uncertain parameters. Typically, a large scenario set is often required to make the distribution of scenarios $\widetilde{\mathcal{P}}$ well fit $\mathcal{P}$, which may cause intractable MIP for a given solver. In this paper, we aim to find a small number of informative representatives in $\{\omega_i\}_{i=1}^N$ by learning representations of scenarios, which is supposed to considerably boost the computation efficiency with tolerable approximation errors.

### 3.2 CONDITIONAL VARIATIONAL AUTOENCODER

As an unsupervised generative model, CVAE is developed on top of VAE, and it further controls the data generation process conditioned on additional random variables (Sohn et al., 2015). These conditional variables could be either class labels or certain properties of the data with specific distributions, which are engaged in the input to both the encoder and decoder. Optimized with the stochastic gradient variational bayes (SGVB) framework (Kingma & Welling, 2014), the evidence lower bound objective (ELBO) of CVAE on the marginal likelihood for the input data is defined as:

$$\log p_\theta(X, c) \geq \text{ELBO}(X, c) = \mathbb{E}_{q_\phi(z|X,c)}[\log p_\theta(X|z, c)] - \text{KL}[q_\phi(z|X, c) \| p(z|c)], \tag{4}$$

where $X$, $c$, $z$ denote input, latent and conditional variables, respectively; both the encoder $q_\phi$ and decoder $p_\theta$ are typically parameterized by deep neural networks. Rather than simple distributions of conditional variables in most existing works, in this paper we exploit CVAE to learn scenario representation in SIPs, with conditional variables derived from continuous parameters in the context. For more details of CVAE, we refer readers to (Kingma et al., 2014; Feng et al., 2021).

## 4 METHODOLOGY

Given a class of SIP instances $\{X_m\}_{m=1}^M$ with parameters drawn from a distribution $\mathcal{D}$, we regard each instance as a 2-tuple $X_m = (D_m, \{\omega_m^i\}_{i=1}^N)$, where $D_m$ denotes the *context* (i.e. the group of static parameters) in the $m$-th instance, and $\omega_m^i$ denotes the $i$-th scenario (i.e. the $i$-th realization of uncertain parameters) in the $m$-th instance. We aim to learn latent representations (variables) $\{z_m^i\}_{i=1}^N$ of scenarios $\{\omega_m^i\}_{i=1}^N$ under the context $D_m$ in each instance.

### 4.1 CVAE FOR SCENARIO REPRESENTATION

Our CVAE comprises an encoder for the *inference* process and a decoder for the *generation* process, which are parameterized by deep neural networks $q_\phi$ and $p_\theta$, respectively. In the generation process, the decoder approximates the posterior distribution of uncertain parameters in scenarios, given the *latent* and *conditional* variables, such that:

$$p_\theta(\omega_m^i|z_m^i, c_m) = f(\omega_m^i; z_m^i, c_m, \theta); \quad q_\phi(c_m|D_m) = h(c_m; D_m, \phi), \tag{5}$$

where $f$ and $h$ denote likelihood functions (e.g. Gaussian or multinomial distributions) that are instantiated by $p_\theta$ and $q_\phi$, respectively. Note that we do not take the raw context $D_m$ as conditional variables for the decoder since they are continuous and high-dimensional. It may not only intensify the computational complexity but also overwhelm the latent variable which is typically low-dimensional. In contrast, we leverage the graph neural network (GNN) in the encoder to derive a low-dimensional conditional variable $c_m$ from $D_m$. Further by preserving the marginal independence between latent and conditional variables, the decoder will more effectively govern the generation of $\omega_m^i$ according to the context $D_m$ (Kingma et al., 2014).

In the inference process, the encoder approximates the posterior distribution of latent representations (variables) of scenarios given $(\omega_m^i, D_m)$, which is expressed as:

$$q_\phi(z_m^i|\omega_m^i, D_m) = \mathcal{N}(z_m^i|\mu_\phi(\omega_m^i, D_m), \sigma_\phi^2(\omega_m^i, D_m)), \tag{6}$$

where we assume a Gaussian distribution of scenario representations with $\mu_\phi$ and $\sigma_\phi$ being parameterized by neural networks, respectively. As aforementioned, we exploit a GNN in the encoder to embed the continuous and high-dimensional $\omega_m^i$ and $D_m$, which is also used by the decoder to embed the sole $D_m$. This parameter sharing design contributes to a fast learning of embeddings for deriving the latent and conditional variables, i.e., $z_m^i$ and $c_m$. To learn disentangled representations for these two types of variables, we force the conditional independence as:

$$q_\phi(z_m^i, c_m|\omega_m^i, D_m) = q_\phi(z_m^i|\omega_m^i, D_m)q_\phi(c_m|D_m). \tag{7}$$

In summary, our CVAE hinges on two important steps, i.e., 1) we first use the encoder to attain scenario representations with respect to the context in each SIP instance; 2) we then use the decoder to reconstruct the scenarios from the latent space according to the context.

**Semi-supervised CVAE.** With the inferred scenario representations, we can directly conduct various downstream tasks for solving SIPs, such as scenario reduction using an off-the-shelf clustering algorithm. However, it may ignore the relationship between scenarios and objective values, which is nontrivial since even similar scenarios would deliver different solutions. Therefore, we also extend CVAE to predict the objective function in a semi-supervised manner. Given that the scenarios are embedded into a continuous space by the encoder, the prediction is supposed to be well generalized (Gómez-Bombarelli et al., 2018).

Most CVAE models for semi-supervised learning predict discrete targets and regard the ground truth, e.g., class labels, as conditional variables. In contrast, we predict a continuous property of the data (i.e. the objective value) and directly approximate it through the encoder. To this end, we use

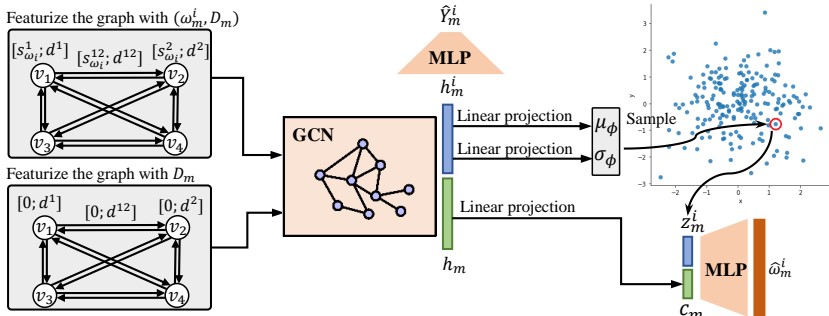

Figure 1: CVAE structure for learning scenario representations

CPLEX solver to collect optimal objective values $\{Y_m^i\}_{i=1}^N$ of the resulting MIP problems, which are defined by respective scenarios with the context, for only a small volume ($1\% \cdot M$) of instances in $\{X_m\}_{m=1}^M$. We refer to the underlying joint distribution of $(\omega_m^i, Y_m^i, D_m)$ as $\mathcal{D}_Y$. Then we infer the objective values with a sub-network, which processes the GNN embedding of $(\omega_m^i, D_m)$ in the encoder, such that the objective function $\sigma$ is parameterized as:

$$r_\psi(Y_m^i|h_m^i) = \sigma(Y_m^i; h_m^i, \psi); \quad q_\phi(h_m^i|\omega_m^i, D_m) = g(h_m^i; \omega_m^i, D_m, \phi), \tag{8}$$

where $h_m^i$ denotes the GNN embedding that is derived by $q_\phi$.

## 4.2 Neural Networks

A major challenge to parameterize CVAE for SIPs is how to derive an effective embedding of the context, since it is high-dimensional and continuous. Moreover, it is also involved in the input to both the encoder and decoder, and significantly influences the eventual scenario representation. A direct use of multi-layer perceptron (MLP) to process static parameters in $D_m$ may fail to leverage the problem structure and cannot generalize across instances of different sizes. To overcome these limits, we exploit a GNN (as aforementioned) to embed the context $D_m$ for attaining conditional variables in the decoder. Meanwhile, we apply the same GNN to embed each scenario with its context $(\omega_m^i, D_m)$ for attaining latent representations in the encoder. To this end, we first describe the context with scenarios on a graph.

**SIP graphs.** We define a complete graph $\mathcal{G} = (\mathcal{V}, \mathcal{E})$, where $\mathcal{V} = \{v_1, \cdots, v_n\}$ denotes nodes with features $\mathbf{V} \in \mathbb{R}^{n \times d_v}$; $\mathcal{E} = \{e_{jk}|v_j, v_k \in \mathcal{V}\}$ denotes edges with features $\mathbf{E} \in \mathbb{R}^{n \times n \times d_e}$. We formally represent the features on the $j$-th node as $\mathbf{v}_{\omega_i}^j = [s_{\omega_i}^j; d^j]$ ([;] means concatenation). In specific, $s_{\omega_i}^j \in \omega_m^i$ denotes a realization of uncertain parameters on the $j$-th node and $d^j \in D_m$ denotes static parameters on the $j$-th node. Similarly, the features on edge $e_{jk}$ is represented as $\mathbf{e}_{\omega_i}^{jk} = [s_{\omega_i}^{jk}; d^{jk}]$, where $s_{\omega_i}^{jk}$ and $d^{jk}$ denote the parameters from the $i$-th scenario and the context, respectively. Different from existing works that only consider varying parameters on nodes, e.g., Nazari et al. (2018), our graph representation is more generic and could be applied to a broader class of SIPs. Particularly, two examples of the graph representation on the network design problem and facility location problem are introduced in Section 4.4.

On top of the SIP graphs, we can exploit GNN to derive embeddings for either the context $D_m$ or varying scenarios with the context, i.e., $(\omega_m^i, D_m)$, $i = \{1, \ldots, N\}$. In our method, we adopt the graph convolutional network (GCN), which is a commonly used GNN variant (Kipf & Welling, 2017; Yao et al., 2019). The architecture of our GCN is similar to the one in Joshi et al. (2019), which was leveraged to compute an adjacency matrix for solving the travelling salesman problem. Specifically, we extend it for graph-based SIP problems to attain graph embeddings for $D_m$ and $(\omega_m^i, D_m)$, denoted as $h_m$ and $h_m^i$, respectively. Detailed structure of the GCN is described in Appendix A. Next, we will revisit our CVAE given the graph embeddings above.

**Encoder.** The encoder processes the graph embedding $h_m^i$ with two separate linear projections to compute 2-dimensional vectors of means and standard deviations for latent variables, respectively, i.e., $\mu_\phi(\omega_m^i, D_m)$ and $\sigma_\phi(\omega_m^i, D_m)$ in Eq. (6). Also, we linearly project the graph embedding $h_m$ into a 2-dimensional vector, i.e., $c_m$ in Eq. (5).

**Decoder.** We concatenate $c_m$ with the latent variables $z_m^i$ and pass them through an MLP to reconstruct the scenario. The MLP is structured by two hidden layers with ReLU activation functions, which are of 128 and 256 dimensions, respectively.

**Semi-supervised learning.** To echo the semi-supervised CVAE in Section 4.1, we leverage another MLP as a sub-network to process $h_m^i$. It comprises a 512-dimensional hidden layer with a ReLU activation function, and outputs a single value to estimate the objective value $Y_m^i$.

The basic structure of neural networks is illustrated in Figure 1, where $\hat{\omega}_m^i$ and $\hat{Y}_m^i$ are the reconstruction and predicted objective value for the $i$-th scenario, respectively. Note that we pad uncertain parameters $\omega_m^i$ with 0 to embed the sole context (in the lower grey square). In this way, the GCN could be shared to attain both graph embeddings $h_m$ and $h_m^i$.

### 4.3 TRAINING & INFERENCE

The training objective is to minimize three loss functions. The first one is defined by the mean square error (MSE) between the reconstructed scenarios from decoder and the original ones. It is essentially employed in CVAE to maximize the marginal likelihood of the data. The second loss is defined by Kullback–Leibler divergence from the latent variables to the prior Gaussian distribution $p(z) = \mathcal{N}(z|0, I)$, which is used for regularization. The third loss is defined by the MSE between the predicted objective values and their ground truth. Formally, the objective for training our CVAE is expressed as:

---

**Algorithm 1** Training procedure

**Input**: encoder $p_\theta$, decoder $q_\phi$, num. of epochs $E$, batch size $B$.

1: Initialize parameters $\theta^{(0)}$ and $\phi^{(0)}$
2: **for** epoch = $1, 2, \ldots, E$ **do**
3:     Shuffle the data in $\mathcal{D}$ and $\mathcal{D}_Y$;
4:     **for** $t = 1, 2, \ldots, \lfloor \frac{|\mathcal{D}|}{B} \rfloor$ **do**
5:         Retrieve a batch of instances from $\mathcal{D}$;
6:         Get a random scenario in each instance;
7:         Compute ELBO in Eq. (9);
8:         $\theta, \phi \leftarrow \text{Adam}(\theta, \phi, \nabla \mathcal{L}(\theta, \phi))$;
9:         **if** objective prediction is trained **then**
10:           Retrieve a batch of instances from $\mathcal{D}_Y$;
11:           Compute MSE of objectives in Eq. (9);
12:           $\phi, \psi \leftarrow \text{Adam}(\phi, \psi, \nabla \mathcal{L}(\phi, \psi))$;
13:         **end if**
14:     **end for**
15: **end for**

---

$$\mathcal{L}(\theta, \phi, \psi) = \mathbb{E}_{\mathcal{D}}[-\text{ELBO}(X_m)] - \alpha \cdot \mathbb{E}_{\mathcal{D}_Y}[\log r_\psi(Y_m^i | h_m^i)], \tag{9}$$

where $\text{ELBO}(X_m) = \mathbb{E}_{q_\phi(z_m^i | \omega_m^i, D_m)}[\log p_\theta(\omega_m^i | z_m^i, c_m)] - \beta \cdot \text{KL}[q_\phi(z_m^i | \omega_m^i, D_m) \| p(z)]$ involves the first MSE and KL divergence; $\alpha$ and $\beta$ are hyperparameters to balance the three losses. Following the typical training paradigm for VAE, we jointly optimize $\theta$ and $\phi$ via reparameterization trick and Monte Carlo approximation (Kingma & Welling, 2014). The training procedure is summarized in Algorithm 1 where we use the Adam optimizer to update the parameters (Kingma & Ba, 2015).

As shown in Algorithm 1, we randomly pick one scenario for each instance so that we can process it with its context in a batch. The resulting generalization across both the contexts and scenarios of different instances will allow for a fast training of neural networks. However, in the inference we employ the trained encoder to attain the latent representations, by processing all scenarios from a set of instances in parallel. It is realized by duplicating the context in each instance for all its scenarios. In doing so, the computational efficiency will be considerably improved for solving the instance with a large cardinality of scenarios, and thus a large amount of instances. For each instance, the derived scenario representations from the encoder are then used for clustering to attain a few centers that correspond to representatives among original scenarios, so we end with reduced scenarios.

### 4.4 APPLICATIONS

We apply our method to learn scenario representation in two typical SIP problems, i.e., the network design problem (NDP) and facility location problem (FLP) (Keutchayan et al., 2021). We represent them as the SIP graphs, whose features on nodes and edges including both static and uncertain parameters are listed in Table 1. For FLP, the parameters tokenized by **F.** and **C.** mean that they are peculiar to facility and customer nodes, respectively. More details about the two problems and their parameters can be found in Appendix B and C.

Table 1: Features on SIP graphs of NDP and FLP

| | Feats | Types | Parameters | Digits |
|---|---|---|---|---|
| **NDP** | $\mathbf{v}_{\omega_i}^j$ | $d^j$ | None | 0 |
| | | $s_{\omega_i}^j$ | demand | 1 |
| | $\mathbf{e}_{\omega_i}^{jk}$ | $d^{jk}$ | opening cost, transportation cost, capacity | 3 |
| | | $s_{\omega_i}^{jk}$ | None | 0 |
| **FLP** | $\mathbf{v}_{\omega_i}^j$ | $d^j$ | coordinate, opening cost (**F.**), capacity (**F.**) | 2(**4**) |
| | | $s_{\omega_i}^j$ | presence (**C.**) | 1 |
| | $\mathbf{e}_{\omega_i}^{jk}$ | $d_{jk}$ | distance between nodes | 1 |
| | | $s_{\omega_i}^{jk}$ | None | 0 |

## 5 EXPERIMENTS

We evaluate our method on NDP and FLP to demonstrate its effectiveness. On the one hand, we train the CVAE to only learn representations for scenarios (Line 5∼8 in Algorithm 1), and the trained encoder can attain representations in each instance, which are directly used for scenario reduction by finding representatives via an off-the-shelf clustering algorithm. On the other hand, we train the semi-supervised CVAE to simultaneously learn scenario representations and predict objective values (Line 5∼13). After the training, the neural network can attain representations and objective values for each scenario and we concatenate them for clustering to find more informative scenarios as representatives. We refer to the above two paradigms as CVAE-SIP and CVAE-SIPA, respectively. In the following experiments, we compare them with other learning based methods, and assess their generalization performance across problems with varying sizes and scenario cardinalities.

### 5.1 SETUP

**Instance generation.** Our instances are generated following the way in (Keutchayan et al., 2021). Specifically, we generate NDP instances with 14 nodes including 2 source nodes, 2 target nodes and 10 transition nodes. For FLP, we designate 10 facility nodes and 20 customer nodes. We generate 200 scenarios for each instance in both problems. To evaluate the generalization performance, we generate larger instances by *doubling* and *quadrupling* the number of *transition* nodes for NDP, respectively. We do the same to both facility and customer nodes for FLP. Besides, we also double and quadruple the scenario cardinality in both problems. More details are provided in Appendix D.

**Training.** We collect $12,800$ instances (of normal size) for NDP and FLP, respectively, and normalize each type of parameters into $[0, 1]$ across instances. Regarding CVAE-SIP, we train our model without the sub-network (used for the prediction task) by solely optimizing the ELBO loss in Eq. (9). Regarding CVAE-SIP, we train neural networks for learning representation and objective prediction simultaneously, as shown in Algorithm 1. We set hyperparameters $\beta = 0.005$ and $\alpha = 100$ in Eq. (9). For both problems, CVAE-SIP and CVAE-SIPA are trained with $100$ and $400$ epochs, respectively. The batch size and learning rate are set to $128$ and $10^{-4}$. We run the training on a server using a single GeForce RTX 2080 Ti GPU.

**Testing.** Pertaining to CVAE-SIP, we employ the $K$-medoids algorithm (Park & Jun, 2009) to obtain representative scenarios based on the latent representations. We keep its default settings and only change the number of clusters (i.e. $K$). Given the attained $K$ representatives, we can derive the approximation solution by reformulating Eq. (3) as:

$$\widetilde{\mathcal{O}}(x) := \min_x \quad \mu^\top x + \frac{N_k}{N}\sum\nolimits_{k=1}^{K} Q(x, \omega_k), \tag{10}$$

where $\omega_k$ means the representative from the k-th cluster and the number of intra-cluster population equals to $N_k$. Accordingly, the approximation error is defined as the gap[1] between objective values of the approximate solution derived from Eq. (10) and the optimal solution from Eq. (3). Pertaining to CVAE-SIPA, we infer both the latent representations and objective values of scenarios, which are concatenated as augmented representations to attain the representatives. To reduce the influence by the potential concurrent processes, we conduct the evaluation on a different desktop computer, which uses a single Nvidia Quadro P4000 GPU with a Xeon W-2133 CPU@3.60 GHz.

### 5.2 COMPARISON ANALYSIS

We compare with four baselines[2], i.e., 1) CPLEX IBM (2017), with default settings to compute optimal solutions. It is also used to solve the approximation problem with representative scenarios. 2) $K$-medoids, the same clustering algorithm as used in our method. However, we apply it to directly cluster the original scenarios rather than the learned low-dimensional representations. 3) Scenario-M, a supervised method with hand-crafted features to predict a scenario for solving a variant of FLP (Bengio et al., 2020). 4) Solution-M[3], a supervised method with raw features to predict the first-stage decision variables for solving blood transshipment problem (Abbasi et al., 2020).

---

[1] In specific, we denote the optimal solutions in Eq. (10) and Eq. (3) by $\tilde{x}$ and $x^*$ respectively. The gap is defined as $|\mathcal{O}(\tilde{x}) - \mathcal{O}(x^*)|/(|\mathcal{O}(x^*)| + \varepsilon)$, where we assume minimization problems and $\varepsilon = 10^{-10}$.

[2] A short comparison with methods in operations research is also given in Appendix I.

[3] The training details of Scenario-M and Solution-M are summarized in Appendix E.

Table 2: Comparison with baselines

| | NDP (14; 200) | | | | | | | | | FLP (30; 200) | | | | | | | | |
| | K=5 | | | K=10 | | | K=20 | | | K=5 | | | K=10 | | | K=20 | | |
| Method | Obj. | Error | Time | Obj. | Error | Time | Obj. | Error | Time | Obj. | Error | Time | Obj. | Error | Time | Obj. | Error | Time |
|---|---|---|---|---|---|---|---|---|---|---|---|---|---|---|---|---|---|---|
| CPLEX | 635.75 | 0.00 | 18s | 635.75 | 0.00 | 18s | 635.75 | 0.00 | 18s | 236.71 | 0.00 | 18s | 236.71 | 0.00 | 18s | 236.71 | 0.00 | 18s |
| Scenario-M | 2533.50 | 2.96 | 0.8s | 2533.50 | 2.96 | 0.8s | 2533.50 | 2.96 | 0.8s | 1980.03 | 7.11 | 0.6s | 1980.03 | 7.11 | 0.6s | 1980.03 | 7.11 | 0.6s |
| Solution-M | 798.81 | 0.26 | 0.8s | 798.81 | 0.26 | 0.8s | 798.81 | 0.26 | 0.8s | 736.21 | 1.87 | 0.2s | 736.21 | 1.87 | 0.2s | 736.21 | 1.87 | 0.2s |
| K-medoids | 976.47 | 0.54 | 0.3s | 761.18 | 0.19 | 0.6s | 677.02 | 0.08 | 1s | 2390.06 | 9.06 | 0.2s | 1344.39 | 4.57 | 0.3s | 502.68 | 1.17 | 1s |
| CVAE-SIP | 930.73 | 0.48 | 0.7s | 734.24 | 0.15 | 0.9s | **637.99** | **0.02** | 1s | 929.61 | 2.87 | 0.7s | 482.70 | 0.99 | 0.7s | 282.49 | 0.23 | 1s |
| CVAE-SIPA | 769.70 | **0.24** | 0.6s | **687.68** | **0.08** | 0.8s | 642.12 | 0.03 | 1s | **709.08** | **1.71** | 0.6s | **381.41** | **0.58** | 0.7s | **264.01** | **0.15** | 1s |

[1] $(n; N)$ means $n$ nodes and $N$ scenarios; **Bold** means the best result from the learning based methods.

Table 3: Generalization to large-scale problems

| | NDP (24; 200) | | | | | | | | | FLP (60; 200) | | | | | | | | |
| | K=5 | | | K=10 | | | K=20 | | | K=5 | | | K=10 | | | K=20 | | |
| Method | Obj. | Error | Time | Obj. | Error | Time | Obj. | Error | Time | Obj. | Error | Time | Obj. | Error | Time | Obj. | Error | Time |
|---|---|---|---|---|---|---|---|---|---|---|---|---|---|---|---|---|---|---|
| CPLEX | 602.47 | 0.00 | 2m | 602.47 | 0.00 | 2m | 602.47 | 0.00 | 2m | 335.37 | 0.00 | 11m | 335.37 | 0.00 | 11m | 335.37 | 0.00 | 11m |
| Scenario-M | 2238.63 | 2.68 | 2s | 2238.63 | 2.68 | 2s | 2238.63 | 2.68 | 2s | - | - | - | - | - | - | - | - | - |
| Solution-M | - | - | - | - | - | - | - | - | - | - | - | - | - | - | - | - | - | - |
| K-medoids | 981.94 | 0.63 | 0.8s | 723.62 | 0.20 | 1s | 662.46 | 0.09 | 3s | 3549.25 | 9.72 | 4s | 1549.53 | 3.74 | 4s | 561.97 | 0.73 | 7s |
| CVAE-SIP | 871.73 | 0.45 | 2s | 696.22 | 0.16 | 2s | 659.92 | 0.09 | 4s | 1083.12 | 2.17 | 3s | 748.59 | 1.21 | 5s | 506.75 | 0.57 | 8s |
| CVAE-SIPA | **809.65** | **0.34** | 2s | **637.72** | **0.06** | 2s | **610.58** | **0.01** | 5s | **974.17** | **1.86** | 3s | **746.02** | **1.18** | 5s | **470.39** | **0.44** | 8s |

| | NDP (44; 200) | | | | | | | | | FLP (120; 200) | | | | | | | | |
| | K=5 | | | K=10 | | | K=20 | | | K=5 | | | K=10 | | | K=20 | | |
| Method | Obj. | Error | Time | Obj. | Error | Time | Obj. | Error | Time | Obj. | Error | Time | Obj. | Error | Time | Obj. | Error | Time |
|---|---|---|---|---|---|---|---|---|---|---|---|---|---|---|---|---|---|---|
| CPLEX | 580.67 | 0.00 | 23m | 580.67 | 0.00 | 23m | 580.67 | 0.00 | 23m | 484.60 | 0.00 | 1h | 484.60 | 0.00 | 1h | 484.60 | 0.00 | 1h |
| Scenario-M | 2163.74 | 2.77 | 3s | 2163.74 | 2.77 | 3s | 2163.74 | 2.77 | 3s | - | - | - | - | - | - | - | - | - |
| Solution-M | - | - | - | - | - | - | - | - | - | - | - | - | - | - | - | - | - | - |
| K-medoids | 803.99 | 0.38 | 3s | 652.78 | 0.12 | 7s | 595.15 | 0.04 | 14s | 6675.13 | 12.56 | 6s | 2803.77 | 4.86 | 21s | 1183.44 | 1.56 | 61s |
| CVAE-SIP | 1079.80 | 0.84 | 5s | **615.99** | **0.06** | 8s | 588.18 | 0.02 | 16s | **1585.53** | **2.27** | 12s | 985.45 | 0.99 | 35s | 627.30 | 0.28 | 56s |
| CVAE-SIPA | **682.35** | **0.17** | 5s | 627.62 | 0.08 | 8s | **581.07** | **0.00** | 17s | 2051.45 | 3.13 | 16s | **910.50** | **0.84** | 33s | **594.33** | **0.22** | 69s |

We evaluate all methods on 200 instances of normal size for NDP and FLP, respectively. We set $K$ = 5, 10, 20 in $K$-medoids for both problems so that we could access the performance against the number of representatives. Note that the computation by CPLEX, Scenario-M and Solution-M is independent of $K$. The results are displayed in Table 2, where we record the average of objective values, approximation errors and computational time over all instances, respectively. As shown, both CVAE-SIP and CVAE-SIPA consistently outperform $K$-medoids in all cases. Though $K$-medoids can also improve the solution quality as $K$ grows, our methods efficiently attain solutions with much smaller errors. It implies that we achieve more informative representatives with the learned low-dimensional representations, which help deliver high-quality solutions. On the other hand, our methods significantly outperform Scenario-M and Solution-M for the three respective values of $K$. This suggests that the representatives found by our methods are able to approximate the original problem more accurately than the approximation with the single scenario or solution. Furthermore, we observe that CVAE-SIPA is generally superior to CVAE-SIP, which indicates that the predicted objective values could effectively enrich the scenario representations for achieving more informative representatives and thus better solutions. Last but not least, all the learning based methods consume fairly short time compared to the sole CPLEX.

## 5.3 GENERALIZATION ACROSS PROBLEM SIZES

A desirable generalization to larger problems is necessary for learning based methods. To verify such ability of our methods, we apply the previously trained networks to directly infer another 50 larger instances (of two levels) for both NDP and FLP, respectively. The instances are generated following the operation of *doubling* and *quadrupling* described in Section 5.1. The baselines are also directly applied to these larger instances except Solution-M since its output dimension (i.e. number of decision variables) is fixed only for the problem of normal size. Similarly, Scenario-M is not applicable to FLP where the scenario dimension (i.e. number of customer nodes) changes with problem sizes. For CPLEX, we set a 1h time limit on FLP (120; 200). In Table 3, the upper and lower half contains the results on *doubling* and *quadrupling* instances, respectively. As shown, CVAE-SIP is superior to $K$-medoids and Scenario-M in most cases except NDP (44; 200) with $K$=5. On the other hand, CVAE-SIPA generally achieves the smallest errors but performs inferior to CVAE-SIP on NDP (44; 200) with $K$=10 and FLP (120; 200) with $K$=5. This might stem from the degenerated accuracy of objective prediction, which induces solutions with relatively large penalties on some instances.

Table 4: Generalization to large scenario cardinalities

| | NDP (14; 400) | | | | | | | | | FLP (30; 400) | | | | | | | | |
|---|---|---|---|---|---|---|---|---|---|---|---|---|---|---|---|---|---|---|
| | K=5 | | | K=10 | | | K=20 | | | K=5 | | | K=10 | | | K=20 | | |
| Method | Obj. | Error | Time | Obj. | Error | Time | Obj. | Error | Time | Obj. | Error | Time | Obj. | Error | Time | Obj. | Error | Time |
| CPLEX | 634.50 | 0.00 | 1m | 634.50 | 0.00 | 1m | 634.50 | 0.00 | 1m | 244.63 | 0.00 | 2m | 244.63 | 0.00 | 2m | 244.63 | 0.00 | 2m |
| Scenario-M | 2266.88 | 2.58 | 1s | 2266.88 | 2.58 | 1s | 2266.88 | 2.58 | 1s | 5260.62 | 12.79 | 2s | 5260.62 | 12.79 | 2s | 5260.62 | 12.79 | 2s |
| Solution-M | **791.95** | **0.26** | 0.7s | 791.95 | 0.26 | 0.7s | 791.95 | 0.26 | 0.7s | 1215.86 | 5.37 | 0.5s | 1215.86 | 5.37 | 0.5s | 1215.86 | 5.37 | 0.5s |
| K-medoids | 972.69 | 0.53 | 0.3s | 730.18 | 0.15 | 0.5s | 650.45 | 0.04 | 1s | 1848.21 | 6.12 | 0.2s | 734.44 | 2.00 | 0.4s | 497.64 | 0.92 | 1s |
| CVAE-SIP | 887.76 | 0.41 | 0.8s | 711.63 | 0.12 | 1s | 656.06 | 0.05 | 2s | **701.73** | **1.83** | 1s | 468.47 | 0.86 | 1s | 439.52 | 0.73 | 2s |
| CVAE-SIPA | 881.05 | 0.38 | 0.8s | **676.99** | **0.06** | 1s | **645.75** | **0.03** | 2s | 723.75 | 1.89 | 1s | **439.08** | **0.77** | 1s | **294.02** | **0.15** | 2s |

| | NDP (14; 800) | | | | | | | | | FLP (30; 800) | | | | | | | | |
|---|---|---|---|---|---|---|---|---|---|---|---|---|---|---|---|---|---|---|
| | K=5 | | | K=10 | | | K=20 | | | K=5 | | | K=10 | | | K=20 | | |
| Method | Obj. | Error | Time | Obj. | Error | Time | Obj. | Error | Time | Obj. | Error | Time | Obj. | Error | Time | Obj. | Error | Time |
| CPLEX | 633.34 | 0.00 | 5m | 633.34 | 0.00 | 5m | 633.34 | 0.00 | 5m | 239.45 | 0.00 | 7m | 239.45 | 0.00 | 7m | 239.45 | 0.00 | 7m |
| Scenario-M | 2137.65 | 2.36 | 0.7s | 2137.65 | 2.36 | 0.7s | 2137.65 | 2.36 | 0.7s | 3807.41 | 9.93 | 2s | 3807.41 | 9.93 | 2s | 3807.41 | 9.93 | 2s |
| Solution-M | 815.13 | 0.28 | 0.8s | 815.13 | 0.28 | 0.8s | 815.13 | 0.28 | 0.8s | 1082.62 | 3.76 | 0.7s | 1082.62 | 3.76 | 0.7s | 1082.62 | 3.76 | 0.7s |
| K-medoids | 896.35 | 0.42 | 0.3s | 770.99 | 0.20 | 0.7s | 678.33 | 0.08 | 1s | 1174.66 | 4.18 | 0.2s | 1091.77 | 3.34 | 0.3s | 732.85 | 1.72 | 1s |
| CVAE-SIP | 900.59 | 0.42 | 1s | **658.40** | **0.03** | 2s | 685.74 | 0.09 | 2s | **600.28** | **1.36** | 2s | **351.18** | **0.49** | 2s | 308.24 | 0.23 | 2s |
| CVAE-SIPA | **761.84** | **0.20** | 1s | 684.33 | 0.07 | 2s | **652.69** | **0.04** | 2s | 990.76 | 2.91 | 2s | 420.95 | 0.77 | 2s | **296.98** | **0.21** | 2s |

Notably, the computational time by CPLEX rises drastically on larger problems, in comparison with those in Table 2. Such low efficiency might not be acceptable in practice, especially where real-time decisions are required for handling a number of problem instances simultaneously. In contrast, our methods deliver solutions with small penalties in short runtime. In Appendix F, we draw curves of the average error against the number of representative scenarios used in FLP problems.

## 5.4 GENERALIZATION ACROSS SCENARIO CARDINALITIES

From a practical view, the cardinality of scenarios in SIP problems might be changed due to certain reasons, e.g., new observations are added or outdated ones are removed (Issac & Campbell, 2017). Therefore, a learning based method for SIPs should be able to handle this varying dimension via generalization. To this end, we evaluate our methods and baselines on 50 instances of NDP and FLP with *doubling* and *quadrupling* scenarios, respectively. The results in Table 4 show that our methods can still achieve the smallest errors across most settings, except NDP (14; 400) with K=5, where Solution-M performs better. It implies that our methods are less sensitive to the scenario cardinality. Since our methods process all scenarios in parallel, they solve instances with scenarios up to 800 in about 2 seconds, which is significantly faster than CPLEX. Additionally, we also test with scenarios generated from varying distributions and different dependencies on the context. The details are provided in Appendix G.

## 5.5 OBJECTIVE PREDICTION

We further demonstrate the efficacy of the semi-supervised learning for objective prediction. The trained model is directly adopted to infer objective values of scenarios in NDP and FLP instances. The results are gathered in Figure 3 and 4, which show our prediction is fairly accurate for both problems even when generalizing to larger problem sizes or scenario cardinalities. It well verifies the effectiveness of our semi-supervised design, which only needs to solve a small volume (1%) of training instances. In addition, the sample efficiency is also enhanced through the generalization over the latent space. More details can be found in Appendix H.

## 6 CONCLUSIONS AND FUTURE WORK

In this paper, we present a deep learning method to solve graph-based two-stage SIPs. We leverage CVAE to learn latent representations of stochastic scenarios conditioned on the context of a SIP instance. With the learned representations, we conduct two downstream tasks for solving SIP, i.e., scenario reduction and objective prediction. The evaluation on two classic SIP problems has shown that our method is able to find less yet informative representatives for improving both solution quality and computation efficiency. Additional studies also verify that our method can be well generalized to larger problem sizes, larger cardinalities of scenarios and different distributions of scenarios. In the future, we will attempt to automate the decision on the optimal number of representatives for scenario reduction, and extend our method towards solving general two-stage SIPs.

## ACKNOWLEDGEMENTS

This research was conducted at Singtel Cognitive and Artificial Intelligence Lab for Enterprises (SCALE@NTU), which is a collaboration between Singapore Telecommunications Limited (Singtel) and Nanyang Technological University (NTU) that is supported by A*STAR under its Industry Alignment Fund (LOA Award number: I1701E0013). Wen Song is supported by the National Natural Science Foundation of China under Grant 62102228, and the Shandong Provincial Natural Science Foundation under Grant ZR2021QF063.

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

# A  GRAPH CONVOLUTIONAL NETWORK

Given a complete SIP graph as defined in Section 4.2, we first process the features of nodes and edges by separate linear transformations as below:

$$\mathbf{V}^0 = \mathbf{V}\mathbf{W}_v^0 + \mathbf{b}_v; \quad \mathbf{E}^0 = \mathbf{E}\mathbf{W}_e^0 + \mathbf{b}_e, \tag{11}$$

where $\mathbf{V} \in \mathbb{R}^{n \times d_v}$ and $\mathbf{E} \in \mathbb{R}^{n \times n \times d_e}$ are the raw features (i.e. parameters in Table 1) on nodes and edges, respectively; $\mathbf{W}_v^0 \in \mathbb{R}^{d_v \times 128}$ and $\mathbf{W}_e^0 \in \mathbb{R}^{d_e \times 128}$ are trainable weights with corresponding bias vectors $\mathbf{b}_v \in \mathbb{R}^{1 \times 128}$ and $\mathbf{b}_e \in \mathbb{R}^{1 \times 128}$. Note that we also use different linear transformations for heterogeneous nodes or edges. For example, the location nodes of facilities and customers are processed separately.

Next, we advance the node and edge embeddings by the graph convolution layer for $L$ iterations, and each iteration has its own parameters, which are expressed as:

$$\mathbf{v}_j^{l+1} = \mathbf{v}_j^l + \text{PReLU}\Big(\text{BN}\big(\mathbf{v}_j^l \mathbf{W}_1^l + \sum_k^{\mathbf{e}_{jk} \in \mathcal{V}} \eta_{jk}^l \odot \mathbf{v}_j^l \mathbf{W}_2^l\big)\Big), \quad \eta_{jk}^l = \frac{\sigma(\mathbf{e}_{jk}^l)}{\sum_k^{\mathbf{e}_{jk} \in \mathcal{V}} \sigma(\mathbf{e}_{jk}^l) + \epsilon} \tag{12}$$
$$\mathbf{e}_{jk}^{l+1} = \mathbf{e}_{jk}^l + \text{PReLU}\Big(\text{BN}\big(\mathbf{e}_{jk}^l \mathbf{W}_3^l + \mathbf{v}_j^l \mathbf{W}_4^l + \mathbf{v}_k^l \mathbf{W}_5^l\big)\Big), \quad l = 0, \ldots, L-1$$

where $\mathbf{W}_1^l, \ldots, \mathbf{W}_5^l \in \mathbb{R}^{128 \times 128}$; $\odot$ means the element-wise product; $\sigma$ is the sigmoid activation function; $\epsilon$ is a small value to avoid numerical error. Initially, the node embedding $\mathbf{v}_j^0$ is the $j$-th row of matrix $\mathbf{V}^0$ and edge embedding $\mathbf{e}_{jk}^0$ is the element at the $j$-th row and $k$-th column of matrix $\mathbf{E}^0$. The above graph layer is similar to the one in Joshi et al. (2019) except that we use parametric rectified linear unit with its learnable parameter being 0.25.

After $L$ ($L$=10) iterations of the above convolution, the node and edge embeddings are represented as $\{\mathbf{v}_j^L\}_{j=1}^n$ and $\{\mathbf{e}_{jk}^L\}_{j,k=1}^n$, respectively. We attain the graph embedding by the mean pooling over all the node embeddings. We neglect edge embeddings considering that their information has already been involved in node embeddings, after multiple convolutions.

**Parameter sharing** The graph embedding network based on Eq. (11) and (12) is able to process the sole context $D_m$ with its static parameters on the graph. Also it can process the context with any scenario in the instance, by featurizing the graph with both static parameters in $D_m$ and a scenario $\omega_m^i$. To foster the learning efficiency, we use a single network to process these two featurizations. It is achieved by padding the uncertain parameters with zeros when we only process the context. We represent the resulting graph embeddings by $h_m$ and $h_m^i$, which are further used to produce conditional and latent variables and meanwhile predict objective values, as stated in Section 4.2.

# B  NETWORK DESIGN PROBLEM

Given a complete directed graph $\mathcal{G} = (\mathcal{V}, \mathcal{E})$, each edge $e_{jk} \in \mathcal{E}$ points from $v_j$ to $v_k$ where $v_j, v_k \in \mathcal{V}$. We denote the set of source and target nodes as $\mathcal{S}$ and $\mathcal{T}$ respectively. The features on an edge include its capacity $q_{jk}$, opening cost $o_{jk}$ and transportation cost $h_{jkc}$ for one unit of commodity $c \in C$.

**Scenarios** The uncertainty exists in the demands on each node. In specific, the parameter $d_{vc}^i$ denotes the demand of commodity $c$ on node $v$, in the $i$-th scenario. A penalty $f_s$ is given to a source node if its commodities are not fully transported in a scenario due to the capacity limit of opened edges.

**Decision variables** The decision variable in the first stage is denoted by $x_{jk}$. $x_{jk} = 1$ means that the edge $e_{jk}$ is open to transport commodities, otherwise $x_{jk} = 0$. The decision variable in the second stage is denoted by $y_{jkc}^i$, which means the units of commodity $c$ transported along the edge $e_{jk}$ in the $i$-th scenario.

The objective of NDP is to decide the edges to be opened, which minimizes the total opening cost and expected transportation cost (including the penalty). The SIP formulation is expressed as:

$$\min \quad \sum_{e_{jk}\in\mathcal{E}} o_{jk}x_{jk} + \frac{1}{N}\sum_{i=1}^{N}\left(\sum_{c\in C}\sum_{e_{jk}\in\mathcal{E}} t_{jkc}h_{jkc}^{i} + \sum_{s\in\mathcal{S}} f_s z_{sc}\right) \tag{13}$$

$$\text{s.t.} \quad \sum_{\substack{v_j=v}}^{e_{jk}\in\mathcal{E}} y_{jkc}^{i} - \sum_{\substack{v_k=v}}^{e_{jk}\in\mathcal{E}} y_{jkc}^{i} = d_{vc}^{i}, \qquad\qquad \forall(v,c,i)\in\mathcal{V}\times C\times\{1,\dots,N\} \tag{14}$$

$$\sum_{\substack{v_j=t}}^{e_{jk}\in\mathcal{E}} y_{jkc}^{i} = \sum_{\substack{v_k=s}}^{e_{jk}\in\mathcal{E}} y_{jkc}^{i} = 0, \qquad\qquad \forall(s,t,c,i)\in\mathcal{S}\times\mathcal{T}\times C\times\{1,\dots,N\} \tag{15}$$

$$d_{sc}^{i} \le \sum_{\substack{v_j=s}}^{e_{jk}\in\mathcal{E}} y_{jkc}^{i} + Mz_{sc}, \qquad\qquad \forall(s,c,i)\in\mathcal{S}\times C\times\{1,\dots,N\} \tag{16}$$

$$\sum_{c\in C} y_{jkc}^{i} \le q_{jk}x_{jk}, \qquad\qquad \forall(v_j,v_k,i)\in\mathcal{V}\times\mathcal{V}\times\{1,\dots,N\} \tag{17}$$

$$z_{sc}, x_{jk}\in\{0,1\}; y_{jkc}^{i}\in[0,\infty). \tag{18}$$

where $M$ is a large value which ensures that the source node will not be penalized if all its commodities are transported.

## C  Facility location problem

The facility location problem comprises $F$ and $C$ locations for facilities and customers, respectively. Each facility location is featured by an opening cost $o_f$, $f\in\{1,\dots,F\}$ and a facility has $q$ units of resources that can be provided to customers. We assume that the number of facilities to be opened is not larger than $v$. For each customer $c\in\{1,\dots,C\}$, it consumes $q_{cf}$ units of resources if it is served by the facility at the location $f$, and the induced service cost is denoted by $s_{cf}$.

**Scenarios** The uncertainty exists in the presence of customers. In specific, the parameter $h_c^i$ refers to the presence of the customer $c$ in the $i$-th scenario, i.e., $h_c^i = 1$ means that the customer $c$ is present to be served in scenario $i$, otherwise $h_c^i = 0$. A penalty $b_f$ is given to the facility $f$ in a scenario if one additional unit of resource is needed for it to meet the demand.

**Decision variables** The decision variable in the first stage is denoted by $x_f$. $x_f = 1$ means that a facility is open at location $f$ to provide services, otherwise $x_f = 0$. The second-stage decision variables include $y_{cf}^i$ and $z_f^i$. In specific, $y_{cf}^i = 1$ means that the customer $c$ is served by facility at location $f$ in the $i$-th scenario, otherwise $y_{cf}^i = 0$. $z_f^i$ means the quantity of additional resources required at facility location $f$ to meet the total demand of its customers, in the $i$-th scenario.

The objective is to decide the locations to build the facilities, so as to minimize the total opening cost and the expected service cost including the potential penalty. The two-stage SIP formulation of the facility location problem is expressed as:

$$\min \quad \sum_{f\in F} o_f x_f + \frac{1}{N}\sum_{i=1}^{N}\left(\sum_{c\in C}\sum_{f\in F} s_{cf}y_{cf}^{i} + \sum_{f\in F} b_f z_f^{i}\right) \tag{19}$$

$$\text{s.t.} \quad \sum_{f\in F} x_f \le v, \tag{20}$$

$$\sum_{c\in C} q_{cf}y_{cf}^{i} \le qx_f + z_f^{i}, \qquad\qquad \forall(f,i)\in F\times\{1,\dots,N\} \tag{21}$$

$$z_f^{i} \le Mx_f, \qquad\qquad \forall(f,i)\in F\times\{1,\dots,N\} \tag{22}$$

$$\sum_{f\in F} y_{cf}^{i} = h_c^{i}, \qquad\qquad \forall(c,i)\in C\times\{1,\dots,N\} \tag{23}$$

$$x_f, y_{cf}^{i}\in\{0,1\}; z_f^{i}\in[0,\infty), \tag{24}$$

where $M$ is a large value that constrains $z_f^i$ to 0 if $x_f = 0$, so as to ensure that the facility $f$ will not be penalized if it is not open.

## D    DETAILS OF INSTANCE GENERATION

We generate NDP instances with 14 nodes, in which the commodities on two source nodes are transported to another two target nodes to meet their demands. The remaining nodes are only for transition without demands. We set two types of commodities and their quantities are uniformly sampled from $[5, 15]$ in each scenario. For each edge, we uniformly sample the opening cost, shipping cost and capacity from $[3, 11]$, $[5, 11]$ and $[10, 41]$, respectively. A penalty 1000 will be triggered for each source node if its commodities cannot be fully transported due to the capacity limit of opened edges. For FLP, we consider 10 facility nodes and 20 customer nodes, with their coordinates uniformly sampled from the unit square $[0, 1] \times [0, 1]$. The opening cost and capacity for each facility are uniformly sampled from $[40, 81]$ and $[30, 60]$. We generate binary values through Bernoulli sampling for customer nodes to indicate their presence in a scenario, with probabilities uniformly sampled from $[0.4, 0.6]$ for each node. A penalty 1000 is set for each facility node if it cannot meet one unit of demands. We generate 200 scenarios for each instance of both problems. To test the generalization performance, we generate larger instances by *doubling* and *quadrupling* the number of *transition* nodes for NDP, respectively. We do the same to both facility and customer nodes for FLP. Regarding the scenario, we also double and quadruple its cardinality in both problems.

## E    TRAINING DETAILS FOR THE TWO BASELINES

**Scenario-M** This method aims to generate one scenario, which can result in a good approximate solution. It is not easy since for each instance, sufficient scenarios need to be evaluated to attain the target scenario for supervised learning. To avoid much computational burden, Bengio et al. (2020) compute the average of scenarios as the initial one and then iteratively improve it to derive a similar solution as the optimal one. However, the improvement heuristic used cannot guarantee that the attained one exists in the original set of scenarios. To tackle this issue, we adapt the method to track the improvement along existing scenarios. Specifically, we find the 20 scenarios nearest to the average and compute the MIPs defined by each of them with the context. We determine the target scenario by picking the one with the most similar solution to the optimal one. This method could be used for both NDP and FLP.

We conduct the same featurization as described in Appendix A.4 in (Bengio et al., 2020), and use both linear regression (LR) and artificial neural network (ANN) with MSE loss. In addition, we also use the same GCN in our method to directly process raw parameters in SIP instances. Specifically, we only input the context with the average of scenarios to GCN, and the graph embedding is linearly projected into output to estimate scenarios. The reported results in the main paper are delivered by GCN, which we find outperforms LR and ANN in most cases.

**Solution-M** Following (Abbasi et al., 2020), we directly predict the first-stage solution in both NDP and FLP. Despite the machine learning algorithms listed in the paper, we use the same GCN in our method for training. It is more powerful to process the raw parameters under the graph representation. In our cases, the raw parameters are the static parameters in the context for both NDP and FLP. We take them as the input into GCN and the output dimension is same as the number of first-stage decision variables. If there is no static parameters on nodes or edges, we use 0 for them to run GCN as usual.

For both supervised methods, we use 12,800 instances (of normal size) in the training with the number of epochs 100, batch size 64 and learning rate 0.001. We test them with diverse instances as described in corresponding experiments in the main paper.

## F    ADDITIONAL RESULTS ON LARGE FLP

We evaluate our methods on FLP of large sizes to show the trend of the average error as the number of representative scenarios grows. We follow the experimental setting in Section 5.3 and apply the trained model to solve 50 instances from FLP (60; 200) and FLP (120; 200). We set $K$ from 1 to

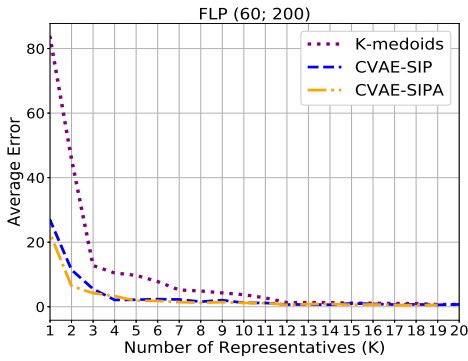
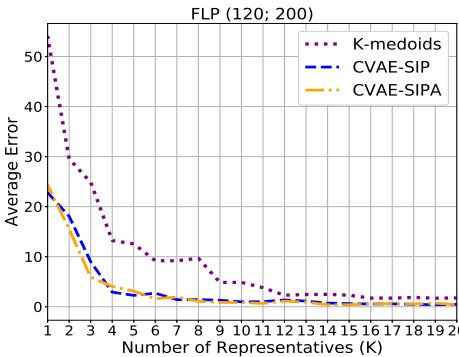

Figure 2: Errors with different values of $K$

20 and thus attain solutions with varying numbers of representative scenarios. The curve of average approximation errors is plotted in Figure 2. We observe that both CVAE-SIP and CVAE-SIPA persistently reduce errors towards 0 as $K$ grows. It means that our methods are consistently attaining more effective representatives from scenarios, which help deliver better solutions. Meanwhile, CVAE-SIP and CVAE-SIPA outperform $K$-medoids with different $K$ on both problems, especially on FLP (120; 200). It again verifies the merit of the learned representations for solving SIP problems. The comparison between CVAE-SIP and CVAE-SIPA implies that though CVAE-SIPA is superior to CVAE-SIP at the early stage of the curve, i.e., $K$=1,...,3, they are on par with each other when more representatives are used eventually.

Table 5: Results on distributions and dependencies

| | NDP (14; 200); Normal | | | | | | | | | NDP (14; 200); Binomial | | | | | | | | |
|---|---|---|---|---|---|---|---|---|---|---|---|---|---|---|---|---|---|---|
| | $K$=3 | | | $K$=5 | | | $K$=10 | | | $K$=3 | | | $K$=5 | | | $K$=10 | | |
| Method | Obj. | Error | Time | Obj. | Error | Time | Obj. | Error | Time | Obj. | Error | Time | Obj. | Error | Time | Obj. | Error | Time |
| CPLEX | 513.82 | 0.00 | 16s | 513.82 | 0.00 | 16s | 513.82 | 0.00 | 16s | 510.08 | 0.00 | 20s | 510.08 | 0.00 | 20s | 510.08 | 0.00 | 20s |
| Scenario-M | 1039.48 | 0.93 | 0.8s | 1039.48 | 0.93 | 0.8s | 1039.48 | 0.93 | 0.8s | 1271.23 | 1.21 | 0.6s | 1271.23 | 1.21 | 0.6s | 1271.23 | 1.21 | 0.6s |
| Solution-M | 678.74 | 0.34 | 0.8s | 678.74 | 0.34 | 0.8s | 678.74 | 0.34 | 0.8s | 729.81 | 0.51 | 0.2s | 729.81 | 0.51 | 0.2s | 729.81 | 0.51 | 0.2s |
| K-medoids | 662.48 | 0.30 | 0.2s | 585.66 | 0.14 | 0.3s | 546.51 | 0.06 | 0.5s | 684.71 | 0.34 | 0.3s | 623.76 | 0.22 | 0.3s | 583.25 | 0.14 | 0.5s |
| CVAE-SIP | 622.98 | 0.23 | 0.6s | **548.45** | **0.07** | 0.6s | **527.59** | **0.03** | 0.9s | 695.60 | 0.36 | 0.6s | **592.40** | **0.16** | 0.6s | 555.35 | 0.09 | 0.9s |
| CVAE-SIPA | **562.87** | **0.11** | 0.5s | 551.94 | 0.07 | 0.6s | 531.94 | 0.04 | 0.8s | **622.91** | **0.22** | 0.5s | 592.91 | 0.16 | 0.6s | **540.54** | **0.06** | 0.9s |

| | NDP (14; 200); D0 | | | | | | | | | NDP (14; 200); D1 | | | | | | | | |
|---|---|---|---|---|---|---|---|---|---|---|---|---|---|---|---|---|---|---|
| | $K$=3 | | | $K$=5 | | | $K$=10 | | | $K$=3 | | | $K$=5 | | | $K$=10 | | |
| Method | Obj. | Error | Time | Obj. | Error | Time | Obj. | Error | Time | Obj. | Error | Time | Obj. | Error | Time | Obj. | Error | Time |
| CPLEX | 563.16 | 0.00 | 17s | 563.16 | 0.00 | 17s | 563.16 | 0.00 | 17s | 617.21 | 0.00 | 17s | 617.21 | 0.00 | 17s | 617.21 | 0.00 | 17s |
| Scenario-M | 1061.20 | 0.87 | 0.7s | 1061.20 | 0.87 | 0.7s | 1061.20 | 0.87 | 0.7s | 1434.12 | 1.37 | 0.9s | 1434.12 | 1.37 | 0.9s | 1434.12 | 1.37 | 0.9s |
| Solution-M | 754.11 | 0.38 | 0.6s | 754.11 | 0.38 | 0.6s | 754.11 | 0.38 | 0.6s | 1282.59 | 1.12 | 0.6s | 1282.59 | 1.12 | 0.6s | 1282.59 | 1.12 | 0.6s |
| K-medoids | 1086.43 | 0.93 | 0.3s | 744.53 | 0.32 | 0.4s | 635.58 | 0.13 | 0.5s | 1091.11 | 0.79 | 0.3s | 816.62 | 0.33 | 0.3s | 692.81 | 0.12 | 0.6s |
| CVAE-SIP | 1070.44 | 0.89 | 0.6s | 698.18 | 0.24 | 0.6s | 632.03 | 0.12 | 0.9s | 1180.53 | 0.90 | 0.6s | 809.02 | 0.31 | 0.6s | 675.19 | 0.09 | 0.9s |
| CVAE-SIPA | **678.80** | **0.20** | 0.5s | **663.97** | **0.18** | 0.6s | **578.57** | **0.03** | 0.8s | **758.14** | **0.23** | 0.5s | **737.81** | **0.19** | 0.5s | **643.50** | **0.04** | 0.8s |

## G  GENERALIZATION ACROSS DISTRIBUTIONS AND DEPENDENCIES

We test the generalization performance of our method on NDP with scenarios sampling from the Normal distribution and Binomial distribution. For the former, we set the expectation and standard deviation to 10 and 2, respectively. For the latter, we set the number of runs and the probability to 20 and 0.5. The results are summarized in the upper half of Table 5. We observe that our methods consistently outperform other learning based approaches with scenarios from both distributions. When $K$=10, both CVAE-SIP and CVAE-SIPA achieve near-optimal solutions with slight approximation errors. On the other hand, it can be found that the results delivered by CVAE-SIPA are generally superior or comparable to those by CVAE-SIP, suggesting a better cross-distribution generalization.

In our training, we do not assume explicit dependencies between the context and scenarios, following the settings in (Keutchayan et al., 2020; Nair et al., 2018; Abbasi et al., 2020). However, we would like to show that the trained networks can well generalize to the dependent settings. In specific, we construct in NDP two dependencies between the demand scenarios and costs in contexts as follows:

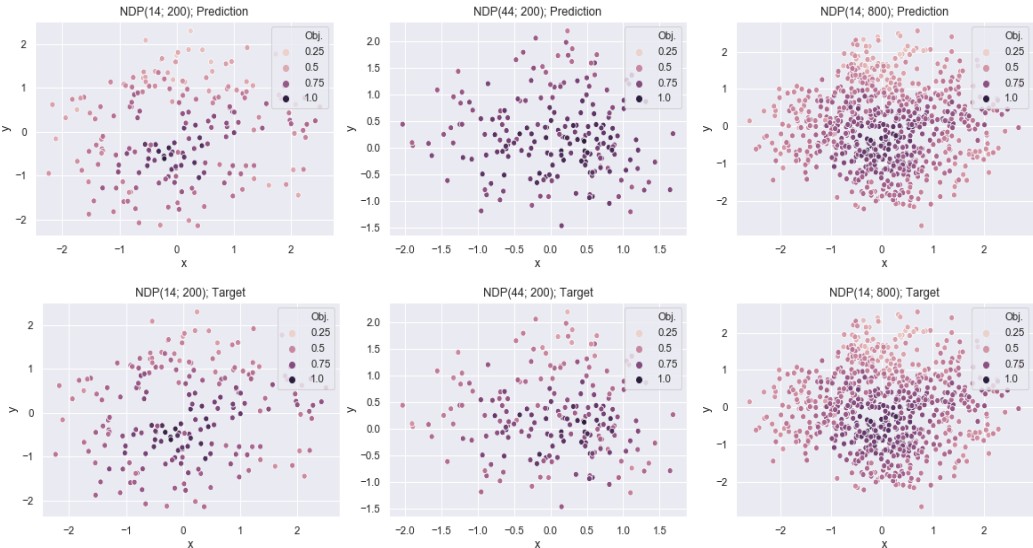

Figure 3: Objective prediction against latent representations (NDP)

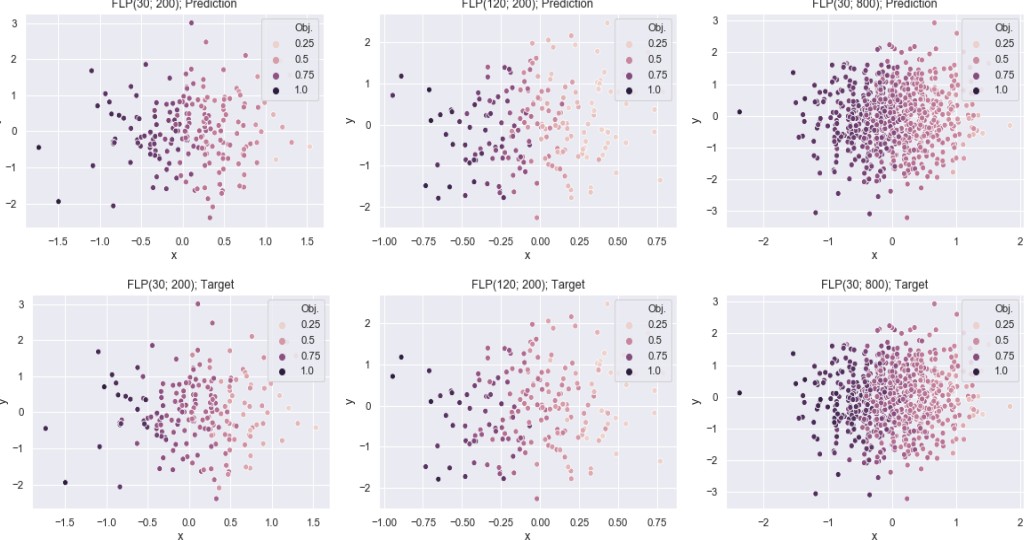

Figure 4: Objective prediction against latent representations (FLP)

1) D0: for each instance, the lower bound of Uniform demands is determined by the smaller value of the average opening and shipping cost in the context; 2) D1: the lower and upper bounds of Uniform demands are determined by the smaller and larger value of the average opening and shipping cost, respectively. We generate 50 instances for each dependency. From the lower half of Table 5, we observe that our methods outperform learning based baselines except the cases with $K$=3, where CVAE-SIP loses its advantage over a few baselines. Generally, it indicates that our methods possess fairly good generalization abilities to the setting where scenarios depend on the contexts.

## H    RESULTS ON OBJECTIVE PREDICTION

We first evaluate our trained model for the prediction task in three FLP instances, which are randomly generated from FLP (30; 200), FLP (120; 200) and FLP (30; 800), respectively. For each instance, we draw a scatter plot of the 2-dimensional latent representations for scenarios and represent the normalized objective values using different colors. The results are shown in the upper half

Table 6: Comparison with OR methods

| | NDP (14; 200) | | | | | | | | | FLP (30; 200) | | | | | | | | |
|---|---|---|---|---|---|---|---|---|---|---|---|---|---|---|---|---|---|---|
| | *K*=5 | | | *K*=10 | | | *K*=20 | | | *K*=5 | | | *K*=10 | | | *K*=20 | | |
| Method | Obj. | Error | Time | Obj. | Error | Time | Obj. | Error | Time | Obj. | Error | Time | Obj. | Error | Time | Obj. | Error | Time |
| CPLEX | 635.75 | 0.00 | 18s | 635.75 | 0.00 | 18s | 635.75 | 0.00 | 18s | 236.71 | 0.00 | 18s | 236.71 | 0.00 | 18s | 236.71 | 0.00 | 18s |
| CSSC | **695.08** | **0.10** | 17s | 689.43 | 0.09 | 17s | 669.59 | 0.05 | 30s | **338.84** | **0.39** | 12s | **299.42** | **0.23** | 12s | 264.98 | 0.16 | 14s |
| Scen-red | 951.54 | 0.50 | 0.3s | 739.41 | 0.18 | 0.5s | 650.80 | 0.03 | 1s | 640.07 | 1.54 | 0.3s | 323.34 | 0.29 | 0.3s | **246.57** | **0.08** | 1s |
| CVAE-SIP | 930.73 | 0.48 | 0.7s | 734.24 | 0.15 | 0.9s | **637.99** | **0.02** | 1s | 929.61 | 2.87 | 0.7s | 482.70 | 0.99 | 0.7s | 282.49 | 0.23 | 1s |
| CVAE-SIPA | 769.70 | 0.24 | 0.6s | **687.68** | **0.08** | 0.8s | 642.12 | 0.03 | 1s | 709.08 | 1.71 | 0.6s | 381.41 | 0.58 | 0.7s | 264.01 | 0.15 | 1s |

| | NDP (44; 200) | | | | | | | | | FLP (120; 200) | | | | | | | | |
|---|---|---|---|---|---|---|---|---|---|---|---|---|---|---|---|---|---|---|
| | *K*=5 | | | *K*=10 | | | *K*=20 | | | *K*=5 | | | *K*=10 | | | *K*=20 | | |
| Method | Obj. | Error | Time | Obj. | Error | Time | Obj. | Error | Time | Obj. | Error | Time | Obj. | Error | Time | Obj. | Error | Time |
| CPLEX | 580.67 | 0.00 | 23m | 580.67 | 0.00 | 23m | 580.67 | 0.00 | 23m | 484.60 | 0.00 | 1h | 484.60 | 0.00 | 1h | 484.60 | 0.00 | 1h |
| CSSC | **618.10** | **0.07** | 2m | 617.94 | 0.07 | 2m | 612.77 | 0.06 | 4m | 1101.83 | 1.28 | 34s | 844.24 | 0.76 | 42s | 715.46 | 0.59 | 2m |
| Scen-red | 894.90 | 0.57 | 3s | 667.90 | 0.15 | 6s | 616.23 | 0.07 | 15s | **1056.69** | 1.19 | 6s | **793.45** | **0.68** | 15s | 607.69 | 0.26 | 60s |
| CVAE-SIP | 1079.80 | 0.84 | 5s | **615.99** | **0.06** | 8s | 588.18 | 0.02 | 16s | 1585.53 | 2.27 | 12s | 985.45 | 0.99 | 35s | 627.30 | 0.28 | 56s |
| CVAE-SIPA | 682.35 | 0.17 | 5s | 627.62 | 0.08 | 8s | **581.07** | **0.00** | 17s | 2051.45 | 3.13 | 16s | 910.50 | 0.84 | 33s | **594.33** | **0.22** | 69s |

of Figure 4, and the corresponding ground truth (i.e. optimal solution values attained by CPLEX) are displayed in the lower half. It is clear that the prediction is fairly accurate with almost the same patterns as the target values in the ground truth, even for the larger problem size or scenario cardinality. It indicates that our design for semi-supervised learning is effective and the trained model exhibits a desirable generalization ability.

We further evaluate the objective prediction on NDP. Specifically, the trained model is used to predict objective values for the scenarios from three instances, which are randomly generated from NDP (14; 200), NDP (44; 200) and NDP (14; 800), respectively. From the results in Figure 3, we observe that our method can also predict the objective values accurately for NDP. Especially, the pattern of predicted values for NDP (14; 800) is almost indistinguishable from the one by ground truth. It reveals that our design for the semi-supervised learning is effective across distinct SIPs.

# I    COMPARISON WITH OR METHODS

In this section, we compare our methods with two mature methods from operations research (OR), i.e., 1) Scen-red (Heitsch & Römisch, 2003), which seeks a subset of scenarios that is closest to original scenarios in terms of Fortet-Mourier probability metric, and 2) CSSC (Keutchayan et al., 2020), which solves a MILP to derive representative scenarios, each of which estimates the average objective of some other scenarios. They are on behalf of the distribution-oriented and problem-oriented paradigm, respectively. The tested instances are the same as the ones in Section 5.2 and 5.3. From the results in Table 6, we observe that our methods are comparable to or better than the baselines and especially on NDP. While CVAE-SIP and CVAE-SIPA are inferior to CSSC on NDP with $K$=5, they attain the best approximate solutions when $K$=10 and 20. Notably, the errors of our methods are close to 0 on both normal and large instances when $K$=20. For FLP, while our methods are inferior to CSSC and Scen-red on FLP when $K$=5 and 10, CVAE-SIPA surpasses CSSC on FLP (30; 200) and gains the smallest objective value and error on FLP (120; 200), when $K$=20. Since our runtime is shorter than CSSC and similar to Scen-red, the results again verify that our methods are effective to reduce scenarios for good approximations.

# J    ABLATION STUDY

We further conduct an ablation study on NDP to verify the effectiveness of the learned scenario representations, and meanwhile investigate the mixed use of these representations with the original scenarios. In specific, we enrich the original scenario set for clustering in the following ways: 1) CVAE-SIPA+Ori.: we concatenate the predicted objective values with their original scenarios (rather than the learned representations in CVAE-SIPA); 2) CVAE-SIP+Ori.: we concatenate the learned representations with their original scenarios (rather than merely use the learned representations in CVAE-SIP). These two kinds of mixed representations are used for clustering to find representative scenarios, where we set $K$=5, 10, 20. The tested instances are the same as the ones in Section 5.2 and 5.3. As shown in Table 6, both CVAE-SIP and CVAE-SIPA are generally superior to their

Table 7: Ablation study

| | NDP (14; 200) | | | | | | | | | NDP (44; 200) | | | | | | | | |
| | $K=5$ | | | $K=10$ | | | $K=20$ | | | $K=5$ | | | $K=10$ | | | $K=20$ | | |
| Method | Obj. | Error | Time | Obj. | Error | Time | Obj. | Error | Time | Obj. | Error | Time | Obj. | Error | Time | Obj. | Error | Time |
|---|---|---|---|---|---|---|---|---|---|---|---|---|---|---|---|---|---|---|
| CPLEX | 635.75 | 0.00 | 18s | 635.75 | 0.00 | 18s | 635.75 | 0.00 | 18s | 580.67 | 0.00 | 23m | 580.67 | 0.00 | 23m | 580.67 | 0.00 | 23m |
| K-medoids | 976.47 | 0.54 | 0.3s | 761.18 | 0.19 | 0.6s | 677.02 | 0.08 | 1s | 803.99 | 0.38 | 3s | 652.78 | 0.12 | 7s | 595.15 | 0.04 | 14s |
| CVAE-SIPA+Ori. | 917.51 | 0.45 | 0.6s | 741.81 | 0.17 | 0.9s | 699.89 | 0.09 | 1s | 918.16 | 0.57 | 5s | 700.10 | 0.19 | 8s | 619.29 | 0.06 | 18s |
| CVAE-SIPA | **769.70** | **0.24** | 0.6s | **687.68** | **0.08** | 0.8s | **642.12** | **0.03** | 1s | **682.35** | **0.17** | 5s | **627.62** | **0.08** | 8s | **581.07** | **0.00** | 17s |
| CVAE-SIP+Ori. | 1208.19 | 0.93 | 0.6s | 850.85 | 0.34 | 0.8 | 679.07 | 0.06 | 1s | **1056.61** | **0.82** | 5s | 785.60 | 0.32 | 9s | 603.39 | 0.04 | 20s |
| CVAE-SIP | **930.73** | **0.48** | 0.7s | **734.24** | **0.15** | 0.9s | **637.99** | **0.02** | 1s | 1079.80 | 0.84 | 5s | **615.99** | **0.06** | 8s | **588.18** | **0.02** | 16s |

[1] **Bold** means the method outperforms its counterpart.

counterparts. The advantage of CVAE-SIPA over CVAE-SIPA+Ori. indicates that with the same predicted objective values, our learned representations are effective to attain better representatives than the original scenarios. Moreover, CVAE-SIP is superior to CVAE-SIP+Ori. probably because the learned representations are overwhelmed by the original scenarios to some extent. It is also interesting to identify an advantage that CVAE-SIP > K-medoids > CVAE-SIP+Ori, which means the pure use of learned representations or original scenarios is better than the mixed use of both.

