# OpenReview forum: "Learning Scenario Representation for Solving Two-stage Stochastic Integer Programs"
_ICLR.cc/2022/Conference — ICLR 2022 Poster_

### Official Review · Reviewer_2AQy · 2021-10-30

**Correctness:** 3
**Technical Novelty And Significance:** 2
**Empirical Novelty And Significance:** 2
**Recommendation:** 6
**Confidence:** 3

**Main Review:**

$\textbf{Strengths}$
-	This paper tackles a challenging problem. It is significant if one can reduce the number of scenarios required for obtaining a high-quality approximation
-	Using CVAE for generating effective scenarios is an interesting idea.
-	The paper provides detailed experimental settings, which are very helpful.

$\textbf{Weaknesses}$

I have the following questions to which I wish the author could respond in the rebuttal. If I missed something in the paper, I would appreciate it if the authors could point them out.

Main concerns:
-	In my understanding, the best scenarios are those generated from the true distribution P (over the scenarios), and therefore, the CVAE essentially attempts to approximate the true distribution P. In such a sense, if the true distribution P is independent of the context (which is the case in the experiments in this paper), I do not see the rationale for having the scenarios conditioned on the context, which in theory does not provide any statistical evidence. Therefore, the rationale behind CVAE-SIP is not clear to me. If the goal is not to approximate P but to solve the optimization problem, then having the objective values involved as a predicting goal is reasonable; in this case, having the context involved is justified because they can have an impact on the optimization results. Thus, CVAE-SIPA to me is a valid method.
-	While reducing the scenarios from 200 to 10 is promising, the quality of optimization has decreased a little bit. On the other hand, in Figure 2, using K-medoids with K=20 can perfectly recover the original value, which suggests that K-medoids is a decent solution and complex learning methods are not necessary for the considered settings. In addition, I am also wondering the performance under the setting that the 200 scenarios (or random scenarios of a certain number from the true distributions) are directly used as the input of CPLEX. In addition, to justify the performance, it is necessary to provide information about robustness as well as to identify the case where simple methods are not satisfactory (such as larger graphs).

Minor concerns:
-	Given the structure of the proposed CVAE, the generation process takes the input of $z$ and $c$ where $z$ is derived from $w$. This suggests that the proposed method requires us to know a collection of scenarios from the true distribution. If this is the case, it would be better to have a clear problem statement in Sec 3. Based on such understanding, I am wondering about the process of generating scenarios used for getting K representatives - it would be great if codes like Alg 1 was provided.
-	I would assume that the performance is closely related to the number of scenarios used for training, and therefore, it is interesting to examine the performance with different numbers of scenarios (which is fixed as 200 in the paper).
-	The structure of the encoder is not clear to me. The notation q_{\phi} is used to denote two different functions q(z|w,D) and $q(c,D)$. Does that mean they are the same network?
-	It would be better to experimentally justify the choice of the dimension of c and z.
-	It looks to me that the proposed methods are designed for graph-based problems, while two-stage integer programming does not have to be graph problems in general. If this is the case, it would be better to clearly indicate the scope of the considered problem. Before reaching Sec 4.2, I was thinking that the paper could address general settings.
-	The paper introduces CVAE-SIP and CVAE-SIPA in Sec 5 -- after discussing the training methods, so I am wondering if they follow the same training scheme. In particular, it is not clear to me by saying “append objective values to the representations” at the beginning of Sec 5.
-	The approximation error is defined as the gap between the objective values, which is somehow ambiguous unless one has seen the values in the table. It would be better to provide a mathematical characterization.


**Summary Of The Paper:**

This paper studies the problem of generating representative scenarios for two-stage stochastic integer programmings in which the parameters could be either static (referred to as context) or stochastic (forming a space of scenarios). The proposed method leverages conditional variational autoencoder to learn the distribution of scenarios conditioned on the context, where the encoder computes the latent variables (based on the context and scenarios) while the decoder recovers the scenarios. In particular, the functions are parametrized by graph neural networks and multilayer perceptron. Experiments on two concrete problems are performed.

**Summary Of The Review:**

This paper considers an interesting problem, but I am not convinced that the proposed models are warranted in theory. In addition, the practical utility is not significant or clear with the presented experiments, and there are many settings needing justifications. However, I look forward to the authors’ response.

---

> ### Author Response · Authors · 2021-11-18
> **Response to Reviewer 2AQy (about the minor concerns)**
>
> **6. The paper introduces CVAE-SIP and CVAE-SIPA in Sec 5 -- after discussing the training methods, so I am wondering if they follow the same training scheme. In particular, it is not clear to me by saying “append objective values to the representations” at the beginning of Sec 5.**
>
> In our original Algorithm 1, we describe the joint optimization with the three losses, which is used to train CVAE-SIPA. In contrast, CVAE-SIP does not involve the MSE loss for objective prediction in its training, since the trained encoder will not predict objective values for scenario reduction. We clarify this issue in Algorithm 1 in our revised paper, where the training for objective prediction in CVAE-SIPA is branched.
>
> Accordingly, we also refine the description of the first graph in Section 5, where we introduce  CVAE-SIP and CVAE-SIPA, as follows:
>
> “We evaluate our method on NDP and FLP to demonstrate its effectiveness. On the one hand, we train the CVAE to only learn representations for scenarios (Line 5$\sim$8 in Algorithm 1), and the trained encoder can attain representations in each instance, which are directly used for scenario reduction by finding representatives via an off-the-shelf clustering algorithm. On the other hand, we train the semi-supervised CVAE to simultaneously learn scenario representations and predict objective values (Line 5$\sim$13). After the training, the neural network can attain representations and objective values for each scenario and we concatenate them for clustering to find more informative scenarios as representatives. We refer to the above two paradigms as CVAE-SIP and CVAE-SIPA, respectively.”
>
> The description “append objective values to the representations” means that in CVAE-SIPA, we link the learned representation and predicted objective value for each scenario, both derived from the trained network. Then we use a clustering algorithm to find centers of these linked scenario representations in an instance, and thus the corresponding representative ones in the raw scenarios.
>
> **7. The approximation error is defined as the gap between the objective values, which is somehow ambiguous unless one has seen the values in the table. It would be better to provide a mathematical characterization.**
>
> We have added the definition of the gap in the first footnote on page 7 as follows:
>
> “In specific, we denote the optimal solutions in Eq. (10) and Eq. (3) by $\tilde{x}$ and $x^{\ast}$ respectively. The gap is defined as  $|\mathcal{O}{(\tilde{x})}-\mathcal{O}{(x^{\ast})}|/(|\mathcal{O}{(x^{\ast})}|+\varepsilon)$, where we assume minimization problems and $\varepsilon=10^{-10}$.”
>
> where $\mathcal{O}(x)$ is defined as the objective value of the original SIP in Eq. (3).

---

> ### Author Response · Authors · 2021-11-18
> **Response to Reviewer 2AQy (about the minor concerns)**
>
> **1. Given the structure of the proposed CVAE, the generation process takes the input of z and c where z is derived from w. tribution. If this is the case, it would be better to have a clear problem statement in Sec 3. Based on such understanding, I am wondering about the process of generating scenarios used for getting K representatives - it would be great if codes like Alg 1 was provided.**
>
> As stated in Section 3 in the original paper, we have defined the distribution of uncertain parameters in a 2-stage SIP as $\mathcal{P}$, which can be continuous or discrete with infinite support. Then we formulate the sample average approximation (SAA) problem of SIP by sampling a set of i.i.d scenarios from P as in Eq. (4). To clarify the problem statement, we follow the reviewer’s advice and define the distribution of scenarios as $\widetilde{\mathcal{P}}$ in Section 3 to differentiate from the true distribution $\mathcal{P}$.
>
> In this paper, we sample a large number of scenarios (e.g. 200) for each instance from its scenario distribution, so as to formulate the scenario set. During training in Algorithm 1, we do not sample additional scenarios. We randomly sample one scenario for each instance from its scenario set, and combine these context-scenario pairs as a batch to compute the loss. We also do not involve the process of getting K representatives, which is only used for testing. During testing, we use the trained encoder to obtain the latent representations of scenarios, which are used for clustering to obtain K centers that correspond to the K representatives. In the revised paper, we have refined the description in Section 3 to clarify how the original problem in Eq. (4) is defined and the description in the last paragraph in Section 4.3 to clarify how to infer the representatives.
>
> **2. I would assume that the performance is closely related to the number of scenarios used for training, and therefore, it is interesting to examine the performance with different numbers of scenarios (which is fixed as 200 in the paper).**
>
> We would like to note that we have conducted this experiment with detailed analysis in Section 5.4 in the original paper, where we generalize our methods to instances with 400 and 800 scenarios respectively, on both NDP and FLP. Please refer to the section for more details.
>
> **3. The notation $q_{\phi}$ is used to denote two different functions q(z|w,D) and q(c|D). Does that mean they are the same network?**
>
> Yes. We use the same GCN to process both the sole context and context-scenario pairs in each instance, to attain the embeddings $h_m$ and $h_m^i$. $h_m$ and $h_m^i$ are linearly projected into $c_m$ and $z_m^i$, respectively. More details of the GCN sharing scheme can be found in Appendix A.
>
> **4. It would be better to experimentally justify the choice of the dimension of c and z.**
>
> In this paper, we set both the dimension of c and z to 2 considering the following aspects: 1) to extract useful features by VAE, the learned representations z should generally be of lower dimensions than original scenarios, which is 4 at least in this paper; 2) for the dimension of c, we also set it by 2 to well influence the scenario representation z but not to overwhelm it in the decoding.
>
> We empirically keep the dimensions of c and z unchanged on both NDP and FLP to simplify the implementation, which perform fairly well on both problems. We will follow the reviewer’s advice and update the results later since the models need to be re-trained in that case.
>
> **5.  It would be better to clearly indicate the scope of the considered problem (i.e., graph-based problems).**
>
> We thank the reviewer for this point. We have revised our description throughout the paper to clarify that our scope is focused on graph-based SIPs. Please refer to our revised paper.
>
> We also would like to note that while our method is limited to SIPs represented by graphs, it is already more general than existing learning based models, which are more strictly limited to specific variable types, certain problems, fixed problem sizes, etc. For example, the baselines Scenario-M and Solution-M in our experiments are not scalable to instances with sizes differing from the training ones.

---

> > ### Author Response · Authors · 2021-11-20
> > **Supplementary experiments for the 4th minor concern**
> >
> > We train CVAE-SIP with different dimensions of the latent and conditional variables z and c, to show how they impact on the performance to solve SIPs. Except for the dimension 2 used in our original settings, we set 3 and 4 for both z and c during training, respectively. We test the resulting CVAE-SIP on 50 NDP instances with 14 nodes and 30 larger instances with  44 nodes. The results are displayed in the table below:
> >
> > |   NDP (14; 200)   |            |    K=5   |      |            |   K=10   |      |            |   K=20   |      |
> > |:-----------------:|:----------:|:--------:|:----:|:----------:|:--------:|:----:|:----------:|:--------:|:----:|
> > | Methods           |    Obj.    |   Error  | Time |    Obj.    |   Error  | Time |    Obj.    |   Error  | Time |
> > | CPLEX             |   639.82   |   0.00   |  18s |   639.82   |   0.00   |  18s |   639.82   |   0.00   |  18s |
> > | CVAE-SIP          | **810.41** | **0.27** | 0.6s | **711.43** | **0.11** | 0.7s | **647.63** | **0.01** |  1s  |
> > | CVAE-SIP-3d       |   913.75   |   0.43   | 0.6s |   713.57   |   0.11   | 0.7s |   664.22   |   0.04   |  1s  |
> > | CVAE-SIP-4d       |   845.79   |   0.32   | 0.6s |   778.54   |   0.21   |  1s  |   652.89   |   0.02   |  1s  |
> > | **NDP (44; 200)** |            |    **K=5**   |      |            |   **K=10**   |      |            |   **K=20**   |      |
> > | Methods           |    Obj.    |   Error  | Time |    Obj.    |   Error  | Time |    Obj.    |   Error  | Time |
> > | CPLEX             |   576.77   |   0.00   |  25m |   576.77   |   0.00   |  25m |   576.77   |   0.00   |  25m |
> > | CVAE-SIP          |   932.46   |   0.63   |  5s  | **646.18** | **0.12** |  8s  | **590.97** | **0.03** | 16s  |
> > | CVAE-SIP-3d       | **917.18** | **0.59** |  5s  |   705.34   |   0.22   |  8s  | 604.55     | 0.05     | 16s  |
> > | CVAE-SIP-4d       |    946.6   |   0.64   |  5s  |   710.29   |   0.23   |  9s  | 599.36     | 0.04     | 16s  |
> >
> > As shown, CVAE-SIP with 2-dimensional c and z generally performs better than the ones with dimensions 3 and 4 (i.e. CVAE-SIP-3d and CVAE-SIP-4d), on both classes of instances. While CVAE-SIP-2d and CVAE-SIP-3d can also deliver near-optimal solutions when K=20, our original CVAE-SIP attains the smallest error which is closer to 0. Furthermore, all CVAE-SIP models cost much less time than CPLEX, especially on the large instances (i.e. 5s vs 25m).

---

> > ### Comment · Reviewer_2AQy · 2021-11-22
> > **Response**
> >
> > "We would like to note that we have conducted this experiment with detailed analysis in Section 5.4 in the original paper, where we generalize our methods to instances with 400 and 800 scenarios respectively, on both NDP and FLP. Please refer to the section for more details."
> >
> > My point was to examine different numbers of scenarios for one dataset, e.g., enumerating scenarios from 1 to 200 for NDP. But I understand that it may not be possible to have many new experiments during the response phase. This is just my minor concern.

---

> > > ### Author Response · Authors · 2021-11-23
> > > **Response to Reviewer 2AQy (about the 2nd minor concern)**
> > >
> > > For scenario reduction methods such as ours, the number of scenario representatives is usually a small number (e.g. K=10 in [Keutchayan2021]). Therefore, we choose K from 1 to 20 in our experiments.

---

> ### Author Response · Authors · 2021-11-18
> **Response to Reviewer 2AQy (about the 2nd main concern)**
>
> **2.  While reducing the scenarios from 200 to 10 is promising, the quality of optimization has decreased a little bit. On the other hand, in Figure 2, using K-medoids with K=20 can perfectly recover the original value, which suggests that K-medoids is a decent solution and complex learning methods are not necessary for the considered settings. In addition, I am also wondering the performance under the setting that the 200 scenarios (or random scenarios of a certain number from the true distributions) are directly used as the input of CPLEX. In addition, to justify the performance, it is necessary to provide information about robustness as well as to identify the case where simple methods are not satisfactory (such as larger graphs).**
>
> **Advantage over K-medoids:** In figure 2, the advantage of our methods over K-medoids is dim due to the large range of the error axis. In our revised paper, we have updated the results in the main text with K=20. According to the average objective and error, our methods have a clear advantage over K-medoids across all problems. For example, Table 3 shows that on FLP (120; 200), our method CVAE-SIPA achieves the average objective of 594.33 with an error of 0.22, while those values of K-medoids are 1.56 and 1183.44, respectively. Moreover when K=20, most errors of K-medoids are still larger than 1 on FLP with various sizes and scenario cardinalities (across Table 2-4), while CVAE-SIPA can reduce them to less than 0.44, with almost the same run time.
>
> **Results with 200 scenarios:** Please note that in the original paper, we have already provided the results of CPLEX with 200 scenarios (i.e. the first row in each table). We run CPLEX with a 1h time limit to obtain the optimal solutions, which serve as benchmarks to compute the approximate error of our methods and other baselines. While our methods cannot achieve optimality with scenario reduction, they significantly outperform other approximate baselines, and meanwhile run much faster than CPLEX especially on large-scale problems. In our revised paper, we have updated our main results with K=20 which show that the advantage of our methods is more obvious. For example, Table 3 shows that CVAE-SIPA achieves 0.00 error on NDP (44; 200) when K=20, with significantly shorter time than CPLEX (2s vs 23m). On FLP (120; 200), it achieves the smallest error of 0.22 among learning based methods, with much shorter time than CPLEX (1m vs 1h time limit).
>
> **Information about robustness:** We are not quite sure what robustness refers to here. If it refers to the performance of our methods in different problem settings, we have conducted extensive experiments on two representative problems with different settings (e.g. varying sizes, number of scenarios and representatives, different scenario distributions). Generally, our methods consistently outperforms the learning based baselines in terms of objective values and errors with relatively short runtime, which shows that our methods are fairly robust in different problem settings. If it refers to the randomness of training, we have tried training our models with different random seeds, and the results are quite stable.

---

> > ### Comment · Reviewer_2AQy · 2021-11-22
> > **Response**
> >
> > I am confused about CPLEX. The paper claims that it can produce an optimal solution, but the above response suggests that the results of CPLEX are obtained based on 200 scenarios (meaning that it is an approximation).
> >
> > Robustness could be measured by variance or std of the testing results.

---

> > > ### Author Response · Authors · 2021-11-23
> > > **Response to Reviewer 2AQy (about the 2nd main concern)**
> > >
> > > We apologies for this confusion. By optimal, we mean optimal solution of the SAA problem defined in Eq. (3), which is an approximation of the original SIP defined in Eq. (1) by sampling a large number of scenarios $\widetilde{\mathcal{P}}$ from uncertain parameters $\mathcal{P}$. This is a common transformation for SIPs since the uncertain parameters could be continuous or discrete with infinite support. We will clarify this point in the future version.
> > >
> > > For robustness, our testing results have small variances. We conduct a quick test to solve 30 instances from NDP(14; 200) and FLP(30; 200), respectively. As shown in the table, our methods deliver relatively small standard deviations of errors.
> > >
> > > |   NDP (14; 200)   |   K=5   |       |   K=10   |      |   K=20   |      |
> > > |:-----------------:|:-------:|-------|:--------:|:----:|:--------:|------|
> > > | Methods           | Error   | Std   | Error    | Std  | Error    | Std  |
> > > | K-medoids         |   0.37  |  0.54 |   0.16   | 0.22 |   0.06   | 0.11 |
> > > | CVAE-SIP          |   0.38  |  0.72 |   0.11   | 0.17 |   0.05   | 0.14 |
> > > | CVAE-SIPA         |   0.18  |  0.23 |   0.05   | 0.11 |   0.00   | 0.01 |
> > > | **FLP (30; 200)** | **K=5** |       | **K=10** |      | **K=20** |      |
> > > | Methods           | Error   | Std   | Error    | Std  | Error    | Std  |
> > > | K-medoids         |  10.63  | 13.41 |   3.27   | 5.15 |   1.17   | 3.57 |
> > > | CVAE-SIP          |   1.63  |  2.78 |   0.88   | 0.84 | 0.30     | 0.57 |
> > > | CVAE-SIPA         |   1.98  |  3.06 |   0.44   | 0.98 | 0.22     | 0.64 |

---

> > > > ### Comment · Reviewer_2AQy · 2021-11-23
> > > > **Response**
> > > >
> > > > Thanks for the response. The std looks very high, compared to the mean.
> > > >
> > > > Given the author's other response, I am happy to increase my rate to 5.

---

> > > > > ### Author Response · Authors · 2021-11-24
> > > > > **Response to Reviewer 2AQy (about the 2nd main concern)**
> > > > >
> > > > > We greatly thank the reviewer for raising the score.
> > > > >
> > > > > We would like to note that the standard deviations of our methods are generally smaller than those of K-medoids. The reason why they look high is that we set large penalties (1000) for nodes in both NDP and FLP, which is larger than the optimal solution (in hundreds).  Thus, it leads to the seemingly high fluctuation of the errors. Regarding the influence of penalties, please kindly refer to our response to the 2nd comment from reviewer Qekv.
> > > > >
> > > > > We further provide a quick test on 30 instances on NDP and FLP respectively, where we set the penalties to 100 for the nodes. The results below verify that all standard deviations decrease drastically and ours are significantly smaller than K-medoids, meaning a better robustness.
> > > > >
> > > > > |   NDP (14; 200)   |   K=5   |      |   K=10   |      |   K=20   |       |
> > > > > |:-----------------:|:-------:|------|:--------:|:----:|:--------:|-------|
> > > > > | Methods           | Error   | Std  | Error    | Std  | Error    | Std   |
> > > > > | K-medoids         |   0.05  | 0.08 |   0.01   | 0.03 |   0.003  | 0.007 |
> > > > > | CVAE-SIP          |   0.03  | 0.05 |   0.01   | 0.01 |   0.001  | 0.004 |
> > > > > | CVAE-SIPA         |   0.01  | 0.02 |   0.00   | 0.01 |   0.000  | 0.002 |
> > > > > | **FLP (30; 200)** | **K=5** |      | **K=10** |      | **K=20** |       |
> > > > > | Methods           | Error   | Std  | Error    | Std  | Error    | Std   |
> > > > > | K-medoids         |   0.45  | 0.76 |   0.27   | 0.45 |   0.21   |  0.81 |
> > > > > | CVAE-SIP          |   0.21  | 0.34 |   0.11   | 0.20 | 0.10     | 0.17  |
> > > > > | CVAE-SIPA         |   0.17  | 0.25 |   0.06   | 0.09 | 0.05     | 0.07  |

---

> ### Author Response · Authors · 2021-11-18
> **Response to Reviewer 2AQy (about the 1st main concern)**
>
> We appreciate the reviewer for the valuable comments. We list our response to the concerned issues as follows:
>
> **1.  In my understanding, the best scenarios are those generated from the true distribution P (over the scenarios), and therefore, the CVAE essentially attempts to approximate the true distribution P. In such a sense, if the true distribution P is independent of the context (which is the case in the experiments in this paper), I do not see the rationale for having the scenarios conditioned on the context, which in theory does not provide any statistical evidence. Therefore, the rationale behind CVAE-SIP is not clear to me. If the goal is not to approximate P but to solve the optimization problem, then having the objective values involved as a predicting goal is reasonable; in this case, having the context involved is justified because they can have an impact on the optimization results. Thus, CVAE-SIPA to me is a valid method.**
>
> Our explanations are two-fold:
>
> * We would like to clarify that our intention is not to let the CVAE to approximate the distribution P. As we summarize in the related work about the probability-oriented paradigm for scenario reduction, only focusing on the scenario probability could deliver poor performance. Thus, when the objective prediction is not involved, we train CVAE-SIP to learn low-dimensional embeddings for the pairs of context and scenario, such that similar scenarios with similar contexts could lie closely in the latent space. While the rationale of CVAE is using the output of decoder to approximate the posterior distribution of scenarios, the learned representations from the encoder (i.e. what we use for clustering) are assumed to be Gaussian or Multinomial in prior, and has no direct relation to the original scenario distribution. Instead, the latent space is learned based on the proximity of the context-scenario pairs. In other words, instead of generating scenarios with the decoder, we leverage another main function of VAE models, that is, attaining representations (or feature embeddings) of the input data for downstream tasks using the encoder. We use CVAE so that we can involve context features as well.
>
> * Our experiment setting follows recent works [Keutchayan2021], [Nair2018], [Abbasi2020], where scenarios are sampled from distributions that are independent of contexts. Nevertheless, this does not mean that the learned latent space is only applicable to the independent setting. As mentioned previously, the latent space is learned to reflect the proximity of scenario-context pairs in the input data. Therefore, while there is no strict dependency between scenarios and context in training, the learned latent space could generalize to the dependent setting (essentially more complex data distributions).
>
> To empirically verify the above statement, we have extended Appendix G to include generalization performance of our methods on settings where the scenario distribution depends on the context. While our models are trained without explicit dependencies, experiments show that our models achieve better results than baselines with explicit dependencies. Please refer to our revised paper for more details. We will conduct more experiments of training on settings with explicit distribution-context dependencies, and try to include the results in the final version.
>
> Reference:
>
> [Keutchayan2021] Julien Keutchayan, Janosch Ortmann, and Walter Rei. Problem-driven scenario clustering in stochastic optimization. arXiv preprint arXiv:2106.11717, 2021.
>
> [Nair2018] Vinod Nair, Dj Dvijotham, Iain Dunning, and Oriol Vinyals. Learning fast optimizers for contextual stochastic integer programs. In Proceedings of the 34th Conference on Uncertainty in Artificial Intelligence (UAI). 2018.
>
> [Abbasi2020] Babak Abbasi, Toktam Babaei, Zahra Hosseinifard, Kate Smith-Miles, and Maryam Dehghani. Predicting  solutions  of  large-scale  optimization  problems  via  machine  learning:  A  case  study  in blood supply chain management. Computers & Operations Research. 2020.

---

> > ### Comment · Reviewer_2AQy · 2021-11-22
> > **Response**
> >
> > "such that similar scenarios with similar contexts could lie closely in the latent space." In the case that contexts and scenarios are independent (as in the experiments), similar scenarios do not necessarily (and should not) have similar contexts. That is, CVAE-SIP attempts to build a relationship between independent variables. This is independent of the representation learning technique one uses.

---

> > > ### Author Response · Authors · 2021-11-23
> > > **Response to Reviewer 2AQy (about the 1st main concern)**
> > >
> > > While we did not perform training in the dependent setting, we have empirically verified that our methods can generalize to the dependent settings in Appendix G in the revised paper. In the generalization experiments, CVAE-SIP generally performs better than other baselines since it learns the proximity between context-scenario pairs to some extent. While similar scenarios do not necessarily have similar contexts, what we mean is that neural networks (in the independent setting) have seen a huge volume of context-scenario patterns, and the tested context-scenario pairs similar to any trained pair could lie closely in the latent space.
> > >
> > > In other words, when there is no dependence between contexts and scenarios, CVAE may not be able to learn how the context influences the generated scenario from the decoder. However, CVAE can still learn features of context-scenario pairs. This is also why we can use the learned representations to predict objective values accurately.

---

> ### Author Response · Authors · 2021-12-02
> **Response to Reviewer 2AQy**
>
> We greatly thank the reviewer for increasing the score!

---

### Official Review · Reviewer_Qekv · 2021-11-02

**Correctness:** 3
**Technical Novelty And Significance:** 2
**Empirical Novelty And Significance:** 3
**Recommendation:** 6
**Confidence:** 2

**Main Review:**

AFTER REBUTTAL:

I thank the authors for providing a detailed rebuttal with several additional experiments.

In my opinion, this is a nice paper. The technical novelty w.r.t. the ML community seems a bit limited, however, the empirical results (in particular for K=20) are very strong, showing that the proposed method can lead to  high quality solutions (< 20% optimality gap) while reducing an optimization solver's runtime by several orders of magnitude. I would like to see the paper accepted and have raised my score accordingly.


----------------

The paper is well written with a smooth introduction to SIPs and the related literature towards machine learning for integer programming. Furthermore the numerical results shown are impressive and may suggest tremendous advantage over current methods in scenario decomposition.

I have two major concerns:

1. GVAEs are *very* well studied (e.g., [1]), and a quick search shows there are prior works that explore graph CVAEs (e.g., [2]). However, none of this related literature is discussed, so the placement of this work within the broader GCN literature is unclear and requires further discussion. Further, is there any specific contribution to the GCN literature, or is the proposed method strictly an application of GCNs to a new problem?


2. The numerical results suggest that the proposed method of using k-medoids in a lower dimensional latent space has tremendous advantages over classical k-medoids and sometimes attains $2x$ improvements. However, in some cases (especially for FLP in Table 3 and 4), all of the methods (including the proposed one) generate very poor solutions with $>100%$ optimality gap. For example with the FLPs in Table 4, it would not be practical to use any of the listed scenario reduction techniques if the nominal problem takes only 2-11 minutes and the scenario reduced solutions are so sub-optimal. I expect that we would require more scenarios and I observe  that the appendix includes some results with a more reasonable setting (e.g., Fig 2) with larger $K$. It would be useful to bring those results to the main text. Furthermore, I believe it would be important to expand the tables in the main text to larger $K$ where the conventional scenario reduction baseline (k-medoids) is meaningful. Otherwise, I would be interested to seeing a comparison with more powerful methods from the scenario reduction literature (e.g., [3]), where $K \leq 10$ is justified.


[1] Kipf & Welling, Variational Graph Auto-Encoders (2016)

[2] Chai, Liu, Duffy, & Kim, Learning to Synthesize Cortical Morphological Changes using Graph Conditional Variational Autoencoder (2021)

[3] Rujeerapaiboon, Schindler, Kuh, & Wiesemann, Scenario Reduction Revisted: Fundamental limits and guarantees (2018)




**Summary Of The Paper:**

Solving stochastic integer programs in practice can be difficult for large problems with many scenarios (i.e., sample discretizations of the uncertain distribution), motivating relevant problems such as scenario reduction and objective prediction. This paper explores these problems by using a graph conditional VAE to learn low-dimensional representations of scenarios and perform downstream tasks in this space. The results show improvements over existing scenario reduction and objective prediction methods over the original high-dimensional space.


**Summary Of The Review:**

The improvement over baselines are nice and the paper is well written, but its positioning within the broader literature and the meaningfulness of the numerical results leave fundamental questions that I would like to see answered before improving the score.

---

> ### Author Response · Authors · 2021-11-18
> **Response to Reviewer Qekv**
>
> **3. I expect that we would require more scenarios and I observe that the appendix includes some results with a more reasonable setting (e.g., Fig 2) with larger K. It would be useful to bring those results to the main text. Furthermore, I believe it would be important to expand the tables in the main text to larger  K where the conventional scenario reduction baseline (k-medoids) is meaningful.**
>
> In the original paper, we set K to small numbers since we aim to emphasize our performance on reducing scenarios. We follow the reviewer’s advice and update tables in the main text with K=20. In doing so, we observe that our method is more superior to baselines, and meanwhile keep the advantage of efficiency over CPLEX. For example, errors of CVAE-SIPA for all NDP cases are smaller than 0.05. For most FLP cases, errors are smaller than 0.3. Please refer to the revised paper for detailed results.
>
> **4. Otherwise, I would be interested to seeing a comparison with more powerful methods from the scenario reduction literature (e.g., [3]), where K<=10  is justified.**
>
> We thank the reviewer for providing this additional option to enrich our paper. In the revised paper, we have conducted a short comparison with two representative methods from distribution-oriented and problem-oriented paradigms in the field of operations research (OR). It verifies our superiority to these baselines, especially on NDP. Please note that we put this new experiment in Appendix I, since our main target is still comparing with learning based methods and CPLEX as shown in the main text, which follows the setting for most current learning-to-optimize literature.

---

> ### Author Response · Authors · 2021-11-18
> **Response to Reviewer Qekv**
>
> We appreciate the reviewer for the valuable comment. We have differentiated our scope from graph CVAE literature and updated our main results with K=20. Please refer to our revised paper. The detailed point-to-point responses are as follows:
>
> **1. GVAEs are very well studied (e.g., [1]), and a quick search shows there are prior works that explore graph CVAEs (e.g., [2]). However, none of this related literature is discussed, so the placement of this work within the broader GCN literature is unclear and requires further discussion. Further, is there any specific contribution to the GCN literature, or is the proposed method strictly an application of GCNs to a new problem?**
>
> We would like to note that our motivation is not to propose a VAE-based model for general graph learning like GVAE in [Kipf2016]. Instead, as clarified in the general response, our aim is to propose a novel pipeline that combines GCN and CVAE for solving the complex and challenging SIPs. While we did not make significant changes to the GCN and  CVAE, we managed to come up with a suitable framework that enables learning latent space for scenario reduction and objective prediction, by considering the unique properties of SIPs. In the revised paper, we have moved the details of GCN structure to the appendix to avoid misunderstanding of our scope and contribution, and highlight our CVAE based pipeline for SIPs in the main text. We have also added the discussion and reference (including [Chai2021]) on CVAE for graph learning in the preliminaries of the revised paper as follows:
>
> "We would like to note that while a few CVAE based methods have been applied to specific tasks in deterministic COPs  (Ichteret al., 2018; Hottung et al., 2020) and graph learning (Chai et al., 2021; Yu et al., 2021), this paper instead endeavours to extend CVAE to suitably solve two-stage SIPs."
>
> We also would like to note that authors in [Chai2021] pointed out in the conclusion that they do not use GCN, but merely represent the task as static graphs. In contrast, our task and structure are totally different from prior CVAE literature, with novel components such as SIP graphs and GCN sharing scheme.
>
> **2. In some cases (especially for FLP in Table 3 and 4), all of the methods (including the proposed one) generate very poor solutions with >100 optimality gap. For example with the FLPs in Table 4, it would not be practical to use any of the listed scenario reduction techniques if the nominal problem takes only 2-11 minutes and the scenario reduced solutions are so sub-optimal.**
>
> We use the experiment settings in [Keutchayan2021], which sets large penalties for constraint violations in the second stage. This is reasonable because it is not desirable to break any constraints in practice. As stated in Appendix D, we follow [Keutchayan2021] and set a large penalty of 1000 for a node that cannot meet the constraints in each scenario, which is much larger than optimal objective values. As shown in Table 3, the optimal objective value of FLP(60; 200) is 335.37 on average, while one scenario in an instance will be punished by 1000 if one mere facility node cannot meet one unit of demands from any customer (defined in Appendix C). In other words, if each scenario has one such node, the final objective value in an instance will increase by 1000 which is almost 3 times as much as the optimal objective.
>
> To make our advantage more clear, we follow the advice from the reviewer and update our results and analysis with K=20 in the main text. In doing so, our errors are further closer to 0 for all cases, which are significantly better than baselines. We do not use much larger K since we still aim to reduce scenarios for faster computation.
>
> We also note that the saved runtime is more clear on large-scale problems. On NDP (44; 200), our method achieves an error of 0.00 when K=20, with significantly shorter time than CPLEX (2s vs 23m). On FLP (120; 200), our method achieves an error of 0.22 when K=20, with much shorter time than CPLEX (1m vs 1h (hit time limit)). Please kindly refer to the tables in the main text of the revised paper.
>
>
> Reference:
>
> [Kipf2016] T. N. Kipf and M. Welling. Variational Graph Auto-Encoders. In the 30th International Conference on Neural Information Processing Systems (NIPS). Workshop on Bayesian Deep Learning (NeurIPS-16 BDL). 2016.
>
> [Chai2021] Yaqiong Chai, Mengting Liu, Ben A Duffy, and Hosung Kim. Learning to synthesize cortical morphological changes using graph conditional variational autoencoder. In the 18th International Symposium on Biomedical Imaging (ISBI). 2021.
>
> [Keutchayan2021] Julien  Keutchayan,  Janosch  Ortmann,  and  Walter  Rei. Problem-driven  scenario  clustering  in stochastic optimization. arXiv preprint arXiv:2106.11717, 2021.

---

> ### Author Response · Authors · 2021-11-30
> **Response to Reviewer Qekv**
>
> We greatly thank the reviewer for raising the score!

---

### Official Review · Reviewer_tf4e · 2021-11-04

**Correctness:** 4
**Technical Novelty And Significance:** 2
**Empirical Novelty And Significance:** 3
**Recommendation:** 6
**Confidence:** 3

**Main Review:**

Pros:
* Since the distance in VAE embedding space is expected to reflect the problem domain better than the distance in the input space, the use of the VAE for clustering scenarios is reasonable.
* The paper shows improved performance over 2 recent baselines. While the performance falls short of the CPLEX solver, the proposed approach is generally faster.
* The paper also includes a visualization of the latent space colored by objective prediction scores to motivate the concatenation of the predicted objective to the embeddings towards clustering.

Limitations:
* Technical significance and novelty: The paper contributions are primarily empirical. The conditional VAEs usage is pretty standard - to provide better embeddings for clustering; the paper builds encoder / decoder architectures that are specific to the task using standard building blocks (graph CNNs).
* Given the empirical nature of the paper, I think the paper can be improved with more intuition about what the VAE learns and why it improves over distance alone in this application. The objective prediction visualizations address this to some extent, but this only illustrates why the predicted objective is useful, and one could in principle append this prediction to the raw scenario representation as well. It would also be interesting to know how good are the samples from the VAE, whether one could (successfully) use VAE samples instead/in addition to selected samples from the original set, etc.

Clarity:
* The paper is generally clear and the context is explained well, but the writing can be improved in various places (e.g. missing determinant in "encoder for inference process and a decoder for generation process", "In future").

**Updated after rebuttal**

I think the authors have addressed the concerns I have raised and I have increased my score.

**Summary Of The Paper:**

The paper aims to use deep-learning towards solving generic stochastic integer programs. The proposed approach involves using a conditional VAE to model the distribution over scenarios, and leveraging the scenario's latent space embeddings to enable better representative scenario selection: the embeddings are used as inputs to a standard clustering algorithm that returns representatives.

The paper demonstrates that using VAE embeddings improves performance when compared to using the same clustering algorithm on the raw scenario representations. It further demonstrates that using embeddings concatenated with a predicted objective generally improves performance over using the embeddings alone. Finally, the paper shows improvement over two recent approaches.

**Summary Of The Review:**

The paper demonstrates the use of a conditional VAE in the context of solving stochastic integer programs. The approach seems technically sound and the empirical results show improved performance over both clustering the raw scenarios and over recent baselines. On the other hand (conditional) VAEs have already been used for similar combinatorial optimization tasks before, and their use here towards scenario clustering has small additional significance and novelty.

---

> ### Author Response · Authors · 2021-11-18
> **Response to Reviewer tf4e**
>
> We appreciate the reviewer for the valuable comment. The required experiments are added and please kindly re-evaluate our paper. The detailed point-to-point responses are as follows:
>
> **1. Technical significance and novelty: The paper contributions are primarily empirical. The conditional VAEs usage is pretty standard - to provide better embeddings for clustering; the paper builds encoder / decoder architectures that are specific to the task using standard building blocks (graph CNNs).**
>
> As in the general response, our aim is not to propose a novel neural architecture. Instead, we intend to propose the first CVAE based approach to solve two-stage SIPs. This is technically significant due to two main points: 1) most deep learning methods (including very few CVAEs) for COPs are designed for deterministic problems, which are not applicable when stochastic elements are involved; 2) compared to existing learning based methods for two-stage SIPs, our method is more general without assumption on variable types, problem types, problem sizes, etc. The advantage of our approach is validated in the experiments.
>
> We also notice that current deep learning methods are mostly adapted from standard learning models, such as LSTM [Nazari2018], Transformer [Kool2019], GIN [Zhang2020], CVAE [Hottung2020], etc. While the CVAE used in our approach is standard, the way we apply CVAE is totally different from existing works that employ CVAE for deterministic COPs, which we have clarified in the general response. We have added the CVAE reference for deterministic COPs and the clarification in the introduction and preliminaries in the revised paper, to make our contribution more clear.
>
> Reference:
>
> [Nazari2018] Mohammadreza Nazari, Afshin Oroojlooy, Lawrence Snyder, and Martin Takác. Reinforcement learning for solving the vehicle routing problem. In the 32nd International Conference on Neural Information Processing Systems (NIPS). 2018.
>
> [Kool2019] Wouter Kool, Herke van Hoof, and Max Welling. Attention, learn to solve routing problems!  In the 7th International Conference on Learning Representations (ICLR), 2019.
>
> [Zhang2020] Cong Zhang, Wen Song, Zhiguang Cao, Jie Zhang, Puay Siew Tan, and Xu Chi. Learning to dispatch for job shop scheduling via deep reinforcement learning. In the 34th Conference on Advances in Neural Information Processing Systems (NIPS). 2020.
>
> [Hottung2020] Andŕe Hottung, Bhanu Bhandari, and Kevin Tierney. Learning a latent search space for routing prob-lems using variational autoencoders.  In the 8th International Conference on Learning Representations (ICLR), 2020.
>
> **2. Given the empirical nature of the paper, I think the paper can be improved with more intuition about what the VAE learns and why it improves over distance alone in this application. The objective prediction visualizations address this to some extent, but this only illustrates why the predicted objective is useful, and one could in principle append this prediction to the raw scenario representation as well. It would also be interesting to know how good are the samples from the VAE, whether one could (successfully) use VAE samples instead/in addition to selected samples from the original set, etc.**
>
> We thank the reviewer’s advice and have conducted an ablation study to further verify the effectiveness of the learned scenario representations. The results and analysis are in Appendix J in our revised paper. In summary, we follow the reviewer’s advice and add experiments as below:
>
> **Appending the predicted objective to the raw scenario.**
>
> Results show that when appended to the same predicted objectives, the learned representations deliver significantly better approximation solutions than the raw scenarios, with much smaller errors on all NDP cases. It indicates that the learned representations are more effective in finding representative scenarios.
>
> **Using VAE samples instead/in addition to selected samples from the original set.**
>
> Firstly, we would like to note that the VAE samples have lower dimensions than the raw scenarios, and they cannot be directly used in SIPs. Instead, we find centers in the VAE samples and thus the corresponding representatives in the raw scenarios.
>
> To further show the merit of VAE samples, we follow the reviewer’s advice to concatenate VAE samples with their corresponding raw scenarios. The clustering is conducted on the resulting mixed representations to attain the centers and thus the representatives in the raw scenarios. Results show that the pure use of the learned representations generally achieves better representatives than the mixed ones, with significantly smaller errors. It again verifies the effectiveness of VAE samples for scenario reduction.
>
> **3. The paper is generally clear and the context is explained well, but the writing can be improved in various places.**
>
> We have double checked the paper and refined the writing. Please refer to the revised paper.

---

> ### Author Response · Authors · 2021-12-01
> **Response to Reviewer tf4e**
>
> We greatly thank the reviewer for increasing the score!

---

### Author Response · Authors · 2021-11-18
**General Response to Reviewers**

We greatly appreciate the valuable comments from all reviewers. The additional experiments required by each reviewer is added in the revised paper, and will be explained in detail in the separate response.

In the first place, we would like to clarify the concerns from the reviewers in terms of our novelty, which are from the following perspectives: CVAEs have already been used for combinatorial optimization (reviewer 1); the CVAE usage and building blocks are standard (reviewer 1); our structure is similar to graph VAE (reviewer 2).

* In the literature, we find that only [Ichter2018] and [Hottung2020] have tried to apply CAVEs for solving combinatorial optimization problems (COPs).  However, these methods are designed for deterministic COPs, and are not applicable to solve SIPs. More specifically, CVAEs for deterministic COPs in [Ichter2018], [Hottung2020] were used to encode the solution search space for each instance. But when the uncertainty exists, the landscape of the solution values could be totally different, which makes the learned latent space useless. To the best of our knowledge, our CVAE model is the first one for solving SIPs, which aims to find representative scenarios to reduce the complexity. Moreover, while the above CVAEs for deterministic COPs need (near-)optimal solutions to learn the solution space in a supervised fashion, it is much harder to obtain high-quality training labels for SIPs due to the complexity incurred by the large number of scenarios. In our method, learning is decomposed into unsupervised latent space learning and semi-supervised objective value learning, which is completely different from existing CVAEs for deterministic COPs.
* We would like to clarify that our main contribution is not proposing a completely novel network structure. Instead, we endeavour to construct a suitable pipeline with coordinated components for solving the challenging SIPs, including key designs such as SIP graphs, network sharing scheme, semi-supervised training scheme for CVAE, etc. We also have no intention to propose a CVAE for general graph learning. Instead, we propose a novel way of adopting CVAE to solve SIPs, by considering their unique properties: 1) we represent both instance and scenarios by SIP graphs; 2) we employ CVAE to model the one-to-many relation between instance and scenarios in SIPs. To avoid the vagueness of our contribution, we put details of GCN adaptation in the appendix and emphasize the structure of our CVAE framework in the main text.


Reference:

[Ichter2018] Brian Ichter, James Harrison, and Marco Pavone. Learning sampling distributions for robot motion planning. In the IEEE International Conference on Robotics and Automation (ICRA). 2018.

[Hottung2020] Andr ́e Hottung, Bhanu Bhandari, and Kevin Tierney. Learning a latent search space for routing problems using variational autoencoders. In the 8th International Conference on Learning Representations (ICLR), 2020.

---

### Decision · Program_Chairs · 2022-01-20

**Decision:**

Accept (Poster)

**Comment:**

This paper presents a conditional variational autoencoder (CVAE) approach to solve an instance of stochastic integer program (SIP) using graph convolutional networks. Experiments show that their method achieves high quality solutions with high performance.

It holds merit as an interesting novel application of CVAEs to the ML for combinatorial optimization literature, as well as for the nice empirical results which show a very nice improvement. Two reviewers had a concern that the contribution is a bit narrowly focused toward  MILP-focused journal rather than a general-purpose ML conference since the core contribution is the novel application. On the other hand, they believe that combinatorial optimization has received growing interest from the ML community in recent years.

All three reviewers vote for borderline accept of this paper. The authors have addressed some of reviewers' concerns, hence some reviewers increased their scores throughout the discussion phase.